# Investigation of the occurrence of significant deviations in the magnetopause location: Solar wind and foreshock effects

Niklas Grimmich[1], Adrian Pöppelwerth[1], Martin Owain Archer[2], David Gary Sibeck[3], Ferdinand Plaschke[1], Wenli Mo[4], Vicki Toy-Edens[4], Drew Lawson Turner[4], Hyangpyo Kim[5], and Rumi Nakamura[5]

[1]Institut für Geophysik und Extraterrestrische Physik, Technische Universität Braunschweig, Braunschweig, Germany
[2]Department of Physics, Imperial College London, London, UK
[3]NASA Goddard Space Flight Center, Greenbelt, Maryland, USA
[4]Johns Hopkins University Applied Physics Laboratory, Laurel, Maryland, USA
[5]Space Research Institute, Austrian Academy of Sciences, Graz, Austria

**Correspondence:** Niklas Grimmich (n.grimmich@tu-braunschweig.de)

**Abstract.** Common magnetopause models can predict the location of the magnetopause with respect to upstream conditions from different sets of input parameters, including solar wind pressure and the interplanetary magnetic field. However, recent studies have shown that some effects of upstream conditions may still be poorly understoodsince deviations between models and in situ observations beyond the expected scatter due to constant magnetopause motion are quite common. Using data from the three most recent multi-spacecraft missions to near-Earth space (Cluster, THEMIS and MMS), we investigate the occurrence of these large deviations in observed magnetopause crossings from common empirical models. By comparing the results from different models, we find that the occurrence of these events appears to be model independent, suggesting that some physical processes may be missing from the models. To find these processes, we test whether the deviant magnetopause crossings are statistically associated with foreshocks and/or different solar wind types and show that in at least 40 % of cases the foreshock can be responsible for the large deviations in the magnetopause's location. In the case where the foreshock is unlikely to be responsible, two distinct classes of solar wind are found to occur more frequently in association with the occurrence of magnetopause deviations: the "fast" solar wind and the solar wind plasma associated with transients such as interplanetary coronal mass ejections. Therefore, the plasma conditions associated with these solar wind classes could be responsible for the occurrence of deviant magnetopause observations. Our results may help to develop new and more accurate models of the magnetopause, which will be needed, for example, to accurately interpret the results of the upcoming SMILE mission.

## 1 Introduction

The motion of the magnetopause (MP), the boundary between the Earth's magnetic field and the interplanetary magnetic field (IMF), is driven by pressure variations in the upstream solar wind, changes in the IMF and flow shear between the magnetospheric and shocked solar wind plasma (e.g., Sibeck et al., 1991, 2000; Shue et al., 1997; Plaschke et al., 2009a, b; Dušík et al., 2010; Archer et al., 2024a). On the dayside, the boundary attempts to balance the dynamic, plasma (thermal) and magnetic (from the draped field lines) pressures of the shocked solar wind on the magnetosheath side and the magnetic pressure

on the magnetospheric side, resulting in the MP changing shape and location in response to upstream condition changes but also to some internal processes (e.g., Shue and Chao, 2013; Archer et al., 2024b). Typically, higher solar wind total pressures cause the MP to move closer to Earth than its average position, while lower total pressures allow the magnetosphere to expand.

Under strong southward IMF conditions, magnetic reconnection occurs, where planetary field lines and IMF lines are reconfigured, allowing magnetic flux and energy to be transported around the magnetosphere (Levy et al., 1964; Paschmann et al., 1979, 2013; Petrinec et al., 2022). Due to dayside flux erosion (Aubry et al., 1970; Sibeck et al., 1991; Shue et al., 1997, 1998; Kim et al., 2024) and the transient flux transfer event (Elphic, 1995; Dorelli and Bhattacharjee, 2009; Fear et al., 2017) that result from patchy magnetic reconnection, the MP surface can develop surface waves (Song et al., 1988) and generally moves

earthwards from its nominal position. This is due to a decrease in the magnetic field strength in the dayside magnetosphere because of the transport of flux to the nightside and an increase in the field-aligned current strength (e.g., Maltsev and Liatskii, 1975; Wing et al., 2002; Samsonov et al., 2024). As a result, the magnetic pressure balancing the solar wind pressure is weakened and the MP is pushed inward.

When the IMF is in quasi-radial configuration, i.e. when the IMF cone angle $\vartheta_{\mathrm{cone}}$ between the Earth-Sun line and the IMF

vector $\boldsymbol{B}_{\mathrm{IMF}}$ is less than $30°$ to $45°$, the MP is often found sunward of its nominal position (Fairfield et al., 1990; Merka et al., 2003; Suvorova et al., 2010; Dušík et al., 2010; Samsonov et al., 2012; Park et al., 2016; Grygorov et al., 2017).

The foreshock is formed in an extended region upstream of the bow shock due to a fraction of solar wind particles being reflected at the bow shock and backstreaming along the IMF. As a result, the interaction of solar wind particles with these backstreamed particles excites instabilities and plasma waves (e.g., Eastwood et al., 2005; Wilson, 2016). A foreshock is

present in most IMF configurations, but in the case of radial IMF, the region forms at and in front of the bow shock nose and becomes most important for MP dynamics by modulating the solar wind-magnetosphere interaction, e.g. through the occurrence of foreshock transients (e.g., Sibeck et al., 1999; Turner et al., 2011; Archer et al., 2015b; Grimmich et al., 2024c; Kajdič et al., 2024).

An explanation for the expansion under quasi-radial IMF conditions comes from MHD theory. The reduction and subsequent

redistribution of the total pressure of the solar wind by the bow shock and magnetosheath, results in a lower pressure on the magnetosphere. This stems mainly from the weaker effect of the field line drape, allowing the MP to move outwards to compensate for the pressure changes (see Suvorova et al., 2010; Samsonov et al., 2012).

The development of Kelvin-Helmholtz instabilities (KHIs) due to shear flows across the MP surface is common (but not exclusive) to northward IMF configurations and leads to waves on the magnetospheric flanks that contribute to the MP motion

(Johnson et al., 2014; Kavosi and Raeder, 2015; Nykyri et al., 2017). Other sources of wave activity and oscillation of the MP independent of the KHI include the flux transfer events mentioned above and wave activity within the foreshock region. The upstream waves in the foreshock can convect through the bow shock and magnetosheath to the MP, generating surface waves and also coupling to waves deep in the magnetosphere (Russell et al., 1983; Luhmann et al., 1986; Fairfield et al., 1990; Russell et al., 1997; Plaschke et al., 2013; Petrinec et al., 2022).

In addition to these external conditions and variations that influence the position of the MP, some studies have shown internal processes can play a role in the motion of the MP. For example, Machková et al. (2019) have shown that the accurate

approximation of the terrestrial magnetic field with an eccentric magnetic dipole shifted by about 500 km from the centre which yielded a variation in the magnetopause stand-off distance of 0.2 $R_E$. Effects of magnetosperic currents on MP location have also been reported. The magnetopause distance to the Earth tends to increase for a stronger ring current Machková et al. (e.g., 2019) and decrease with stronger Region 1 Birkeland currents, which accompany the IMF's turning to a southward configuration (e.g., Sibeck et al., 1991).

Empirical models of the MP (e.g. Fairfield, 1971; Formisano et al., 1979; Sibeck et al., 1991; Petrinec and Russell, 1996; Shue et al., 1997, 1998; Boardsen et al., 2000; Chao et al., 2002; Lin et al., 2010; Nguyen et al., 2022a, b) aim to predict the average location of the MP with a given shape under different solar wind conditions. Thus, in response to the upstream conditions described above, global and quasi-static changes in the boundary can be predicted. However, the models cannot capture the more realistic evolution of the boundary under changing conditions, which leads to the constant motion of the MP, resulting in a natural scatter of model predictions compared to spacecraft observations.

Some of these models focus only on specific regions of the magnetosphere, such as Petrinec and Russell (1996) for the near-Earth magnetotail region or Boardsen et al. (2000) for the high latitude regions. The earliest models (e.g., Formisano et al., 1979; Sibeck et al., 1991) use general second order surface polynomials to describe the surface, while conic sections have become the most widely used functional forms for empirical models. The simplest of these widely used models, such as Shue et al. (1997, 1998), assume rotational symmetry that is only influenced by the solar wind dynamic pressure $p_{\mathrm{dyn}}$ and the IMF component $B_z$. More complex models, such as the Lin et al. (2010) or Nguyen et al. (2022a, b) models, assume a more asymmetric shape, including terms describing the indentation of the surface at high latitudes caused by the cusp and taking into account more parameters affecting the shape and location of the MP (e.g., dipole tilt, magnetic pressure, and IMF magnitude).

Although the use of more complex models improves the prediction accuracy for the MP location, all models still have inherent biases and similar errors around 1 $R_E$ (e.g. Šafránková et al., 2002; Case and Wild, 2013; Staples et al., 2020; Aghabozorgi Nafchi et al., 2024). These errors, which are unable to capture the constant motion around the mean location of the magnetopause, could have several causes. On one hand, due to the inherently variable nature and spatial structure of the solar wind, which sometimes shows widely different conditions between measurements a few hundred km apart, the conditions measured at L1 (which are often used for MP modelling) may not affect the Earth as expected (Borovsky, 2018a; Burkholder et al., 2020). Several studies like Walsh et al. (2019), O'Brien et al. (2023) or Aghabozorgi Nafchi et al. (2024) have shown the complications associated with the propagation of the solar wind to Earth. On the other hand, the studies of Grimmich et al. (2023a, 2024b) have identified specific parameters (such as solar wind speed, IMF cone angle, Alfvén Mach number and plasma $\beta$) that seem to be responsible for the deviation of the observed MP crossings from the MP models. It is therefore possible that besides the propagation problem, important mechanisms of the interaction are not yet captured by the models.

For example, high solar wind speeds appear to lead to an anti-Earthward expansion and outward displacement of the MP from the predicted model location based on the assumption that the higher dynamic pressure in these cases compresses the MP (Grimmich et al., 2023a, 2024b). The foreshock is reported to become stronger (more wave activity and increased occurrence rate of transient events) under high solar wind speed conditions (Chu et al., 2017; Vu et al., 2022; Zhang et al., 2022;

Xirogiannopoulou et al., 2024). Thus, one of these overlooked processes could modify of the parameters that affect the MP through the formation of the foreshock region upstream of the Earth's bow shock (e.g., Walsh et al., 2019).

Another point of consideration is the interconnected nature of the solar wind parameters (e.g., Xu and Borovsky, 2015; Borovsky, 2018b). In general, the solar wind plasma can be categorised into several types with systematic differences in solar wind parameters. Most common classification schemes divide the solar wind plasma into three to four main types: coronal hole plasma, ejecta and streamer belt plasma, from which a subset called sector reversal region plasma is sometimes separated (see Xu and Borovsky, 2015; Borovsky, 2020, and references therein). While the ejecta type includes all plasma associated with solar transients such as interplanetary coronal mass ejections (ICMEs), the other three types can be roughly separated in terms of solar wind velocity. The coronal hole plasma is often associated with the "fast" solar wind with velocities around 550 km s$^{-1}$, the streamer belt plasma is often associated with the "slow" solar wind with velocities around 400 km s$^{-1}$, and the sector reversal region plasma is often associated with the "very slow" solar wind with velocities around 300 km s$^{-1}$. The change between the types of solar wind plasmas then causes synchronous changes in the solar wind parameters (Borovsky, 2018b).

Therefore, the approach of Grimmich et al. (2023a) may be too simplistic to identify how deviations from MP models arise by identifying the influence of individual parameters responsible for deviations from the MP models. Since Koller et al. (2024) showed that taking different solar wind types into account improves the classification of magnetosheath ion distributions, it is possible that looking at the response of the MP to different solar wind types will reveal some missing aspects in current models.

Building on the results of previous studies by Grimmich et al. (2023a, 2024b), in this study we investigate the relationship between the observed and the modelled MP deviations in relation to the different solar wind types and quantify the extent to which the foreshock is responsible for the deviations.

## 2  Datasets and Methods

For our investigation we use the Grimmich et al. (2023b, 2024a) and Toy-Edens et al. (2024a) datasets, which contain observation times and locations of magnetopause crossings (MPCs) from the Cluster (Escoubet et al., 2001, 2021), the Time History of Events and Macro-scale Interactions during Substorms (THEMIS, Angelopoulos, 2008) and the Magnetospheric Multiscale (MMS, Burch et al., 2016) missions in the years between 2001 and 2024. For a comprehensive overview of these datasets, we recommend consulting the relevant publications (Grimmich et al., 2023a, 2024b; Toy-Edens et al., 2024b).

Following the previous studies of Grimmich et al. (2023a, 2024b), we also use the high-resolution OMNI data (King and Papitashvili, 2005) of 1 min to associate all crossings in the datasets with upstream conditions. We take averages from an 8-minute OMNI interval preceding each crossing, if no more than 3 data points are missing in that interval for all solar wind parameters. Otherwise, the crossings are not associated with any upstream data. This handling of the upstream data is identical to that used in the previous studies mentioned Grimmich et al. (see, e.g., 2024b, for a detailed discussion). For reference, we also use the 1 min OMNI data from time intervals in the years between 2001 and 2024 where at least one of the three missions is located on the dayside and could potentially observe MPC events.

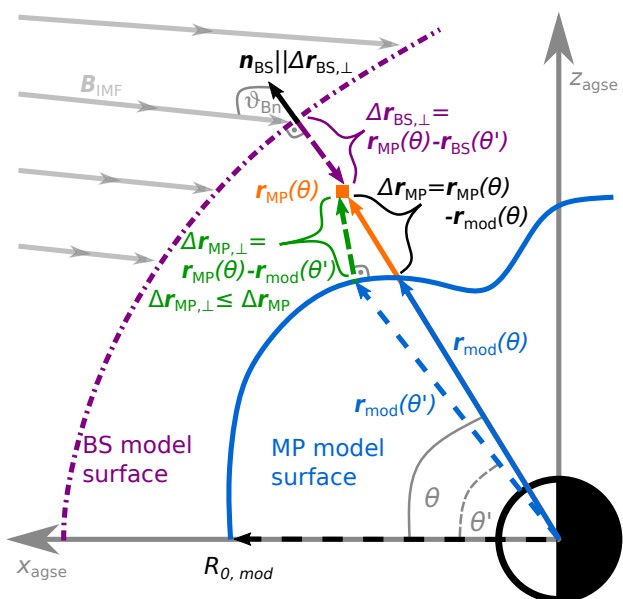

**Figure 1.** Visualisation of the deviation between $r_{\mathrm{MP}}$ and $r_{\mathrm{mod}}$, an observed MP and the modelled MP location. The sketch is simplified, not to scale and shows only the $(x, z)$-plane with an arbitrary (indented) MP model in blue. Including the azimuth angle $\varphi$ next to the zenith angle $\theta$ would lead to the generalisation described in the text. It can be seen that the difference vector $\Delta r_{\mathrm{MP}}$ between the spacecraft observation and the modelled surface of the MP would result in greater distances, when the using $\theta$ (the zenith angle of the spacecraft position) in model calculations. The shortest (minimum) distance along the normal to the surface can be found for the angle $\theta'$ (which can be different from $\theta$) and gives a more physical meaningful representation of the deviation between model and observation (this would also be true for non-indented models). In addition, this sketch shows the geometry of the calculation for the angle $\vartheta_{B,n}$ between $B_{\mathrm{IMF}}$ and the local bow shock normal $n_{\mathrm{BS}}$, estimated as a anti-parallel vector to the smallest distances from the spacecraft position to a modelled bow shock surface.

As the datasets from Grimmich et al. (2023b) and Toy-Edens et al. (2024a) only include dayside events, we have to limit our investigation here to the dayside magnetosphere. Thus, we only use events from the three datasets if they are associated with a positive $x$ component at the observation location in the aberrated Geocentric Solar Ecliptic (aGSE) coordinate system ( e.g., Laundal and Richmond, 2016). The aberration is performed by rotating the coordinates around the $z$-axis with an angle computed individually for each crossing, taking into account the mentioned associated solar wind velocity from the OMNI dataset and the Earth's orbital velocity. For cases where the crossing are not associated with OMNI data we use a mean solar wind velocity of $400 \ \mathrm{km \ s^{-1}}$ for the calulation.

Furthermore, in the Grimmich et al. (2023b, 2024a) datasets, the crossings are associated with a probability value indicating the certainty of the observed MP identification. We follow the recommendation of previous publications and use only the crossings with probabilities above 0.75, which are considered well identified crossings.

Using the associated OMNI data, we calculate the difference between the observed MP position $r_{\mathrm{MP}}$ (i.e. the spacecraft position during a magnetopause crossing) and the position predicted by an MP model $r_{\mathrm{mod}}$, which we can use to identify events

where the model cannot explain the observation. Figure 1 shows a simplified case in the $(x, z)$-plane of the near-Earth space geometry. Here, the simplest approach to calculating the deviation would be to use the zenith angle $\theta$ between $\boldsymbol{r}_{\mathrm{MP}}$ and the $x$ axis to determine the location of the MP model $\boldsymbol{r}_{\mathrm{mod}}$. However, we can see in the sketch that if we consider the zenith angle $\theta'$ for the calculation of $\boldsymbol{r}_{\mathrm{mod}}$, we can determine a perpendicular deviation from the modelled MP surface to the observation point, which is basically the normal displacement of the model MP to the observed crossing. This perpendicular/normal deviation, as seen in the sketch, is shorter than the simple approach calculations and therefore a more physically meaningful and unbiased difference between $\boldsymbol{r}_{\mathrm{MP}}$ and $\boldsymbol{r}_{\mathrm{mod}}$.

To describe this physically meaningful difference in a 3D space, we consider all possible $\theta$ and $\varphi$ angles and take the angles $\theta'$ and $\varphi'$ which yield the smallest deviation from the observed MP location, even though these may not be the angles that describe the observed location. We minimise the term

$$\Delta r = |\boldsymbol{r}_{\mathrm{MP}} - \boldsymbol{r}_{\mathrm{mod}}(\theta, \varphi)|, \tag{1}$$

where $\varphi$ is the azimuth angle between the projection of $\boldsymbol{r}$ in the $(y, z)$ plane and the positive $z$ axis.

By comparing the absolute values of $\boldsymbol{r}_{\mathrm{MP}}$ and $\boldsymbol{r}_{\mathrm{mod}}$, we can also see whether the observed location corresponds to a point further Earthward ($|\boldsymbol{r}_{\mathrm{MP}}| < |\boldsymbol{r}_{\mathrm{mod}}|$) or further anti-Earthward ($|\boldsymbol{r}_{\mathrm{MP}}| > |\boldsymbol{r}_{\mathrm{mod}}|$) than the model prediction. In the following, we refer to the events occurring further Earthward as overestimated MPC and those occurring further anti-Earthward as underestimated MPC. For example, the sketch in Figure 1 illustrates an underestimated MPC.

In this study, we calculate the perpendicular deviation of the MP observation for two different MP models. We use the simple and widely used Shue et al. (1997, 1998) model, hereafter SH98, which describes the MP surface in a rotational symmetry with the function

$$r_{\mathrm{SH98}} = R_{0,\mathrm{SH98}} \left( \frac{2}{1 + \cos\theta} \right)^{\alpha}, \tag{2}$$

where $R_0$ is the magnetopause stand-off distance and $\alpha$ the so-called flaring parameter. Since the SH98 model neglects asymmetries in the MP surface, the influence of the dipole tilt angle $\psi$ on the MP shape, and also the prominent indentation feature associated with the cusp regions of the magnetosphere, we will also present results using the relatively new Nguyen et al. (2022b, a) model, hereafter N22b. This model incorporates the above features, and while similar to the SH98 model in its basic zenith angle function, the IMF dependence in the MP stand-off distance is different, showing a weaker effect on MP motion under changing IMF. The functional form of the N22b model is described by

$$r_{\mathrm{N22b}} = R_{0,\mathrm{N22b}} \left( \frac{2}{1 + \cos\theta} \right)^{\beta} (1 - q(\theta, \varphi, \psi)), \tag{3}$$

where $q(\theta, \varphi, \psi)$ is the term describing the indentation of the surface near the cusp influenced by the dipole tilt angle $\psi$. The flaring parameter of the N22b model $\beta$ is also influenced by the dipole tilt.

Since currents from the inner magnetosphere, such as the ring current, may have an effect on the position of the MP, and thus bias our statistics, we have tried to eliminate these influences from our data. We use the caculation presented in Machková et al. (2019) to modify the model output $r_{\mathrm{mod}}$ with the pressure corrected Disturbance Storm-Time Index $D_{\mathrm{st}}$ (Burton et al., 1975;

Nose et al., 2015). The $D_{st}$ gives information on geomagnetic activity associated with horizontal magnetic field deviations, and can indicate ring current activity and thus be used to limit the effects of the ring current.

We can use the minimisation of (1) with the vector $\boldsymbol{r}_{BS}$ pointing to the surface of a bow shock model (instead of the vector pointing to an MP model surface) to obtain an estimate of the bow shock normal in the vicinity of the MPC observation. The vector $\Delta\boldsymbol{r}_{\perp} = \boldsymbol{r}_{MP} - \boldsymbol{r}_{BS}(\theta', \varphi')$ for the optimal combination of $(\theta', \varphi')$ from this new minimisation should then be roughly aligned with the normal of the model bow shock surface upstream of the MPC observation. Here we use the Chao et al. (2002) model (CH02) and therefore define the bow shock model normal associated with each MPC as

$$n_{BS} = \frac{\boldsymbol{r}_{MP} - \boldsymbol{r}_{CH02}(\theta', \varphi')}{|\boldsymbol{r}_{MP} - \boldsymbol{r}_{CH02}(\theta', \varphi')|}. \tag{4}$$

We choose the sign of $n_{BS}$ so that $n_{BS}$ points to the sun (i.e. the $x$ component is always positive).

The angle $\vartheta_{B,n}$ is defined as the angle between the computed normals $n_{BS}$ and the IMF vector $\boldsymbol{B}_{IMF}$ associated with the magnetopause observation. Typically, the region upstream of the quasi-parallel bow shock ($\vartheta_{B,n} < 45°$) is associated with the foreshock, while no foreshock activity is expected upstream of the quasi-perpendicular bow shock ($\vartheta_{B,n} > 45°$). However, the foreshock also extends into the quasi-perpendicular region, and in some cases $\vartheta_{B,n} < 60°$ is used to define the boundary of the active foreshock region (Wilson, 2016; Karlsson et al., 2021). The angle $\vartheta_{B,n}$ can therefore be used to estimate whether the crossing is observed behind the quasi-parallel or quasi-perpendicular bow shock, and thus be associated with foreshock activity.

An alternative to determine whether the spacecraft observation is behind a quasi-parallel or quasi-perpendicular shock is given in Petrinec et al. (2022). The authors take advantage of the fact that the IMF cone angle is identical to $\vartheta_{B,n}$ in the sub-solar magnetosphere, and map this cone angle to other regions with the clock angle separation between the spacecraft location and the IMF to the corresponding position of the observation. Explicitly, their parameter "$q$" is calculated with

$$q = \cos\left(\vartheta_{cone}\right)\cos\left(\vartheta_{clock,IMF} - \vartheta_{clock,sc}\right) = \frac{B_x}{|B_{IMF}|} \cdot \cos\left(\arctan 2\left(\frac{B_y}{B_z}\right) - \arctan 2\left(\frac{y_{sc}}{z_{sc}}\right)\right), \tag{5}$$

where $B_x$, $B_y$, $B_z$ are the components of $\boldsymbol{B}_{IMF}$ and $y_{sc}$ and $z_{sc}$ are the spacecraft position components. This parameter gives a value between -1 and 1, where $q < 0$ indicates that the spacecraft is behind the quasi-perpendicular shock and $q > 0$ indicates that it is behind the quasi-parallel shock. Both methods are in agreement in up to 80 % of our cases.

We are aware of the fact that this estimate of $\vartheta_{B,n}$ certainly does not give the angle at the bow shock from which the plasma came to influence the MP. Therefore, we use our estimates together with the $q$ value results from eq. (5), and when filtering for shock activity, we use only the result from our datasets where both estimates indicate the desired condition. This approach should minimise errors arising from normal and subsequent angle calculations. Taken together, the MPCs in this study are expected to be observed behind the quasi-parallel bow shock and thus in the foreshock region when $\vartheta_{B,n} < 45°$ and $q > 0$.

In addition, we use the classification scheme introduced in Xu and Borovsky (2015) using the IMF magnitude $|\boldsymbol{B}_{IMF}|$, the solar wind ion velocity $|\boldsymbol{u}_{sw}|$, the solar wind ion density $n_{ion}$ and the solar wind ion temperature $T_{ion}$ to group the crossings according to the different solar wind types with empirically determined thresholds: ejecta (EJC), coronal hole origin (CHO), streamer belt origin (SBO), and sector reversal region (SRR) (see Xu and Borovsky, 2015).

**Table 1.** Median values with median absolute deviation values of different solar wind plasma parameters in the four classes of solar wind plasma from Xu and Borovsky (2015). The parameter values are extracted from the OMNI dataset between the years 2001 and 2024.

|  | All sources | CHO plasma | SBO plasma | SRR plasma | EJC plasma |
|---|---|---|---|---|---|
| $|\boldsymbol{B}_{\mathrm{IMF}}|$ in nT | $4.8 \pm 1.4$ | $5.2 \pm 1.2$ | $4.7 \pm 1.2$ | $3.8 \pm 1.0$ | $9.5 \pm 2.3$ |
| $|\boldsymbol{u}_{\mathrm{sw}}|$ in km s$^{-1}$ | $408.2 \pm 63.1$ | $558.3 \pm 52.7$ | $408.9 \pm 35.6$ | $333.2 \pm 23.2$ | $403.5 \pm 49.2$ |
| $\boldsymbol{n}_{\mathrm{ion}}$ in cm$^{-3}$ | $4.8 \pm 2.0$ | $2.8 \pm 0.8$ | $4.6 \pm 1.4$ | $8.4 \pm 2.7$ | $5.1 \pm 2.6$ |
| $\boldsymbol{T}_{\mathrm{ion}}$ in $10^4$ K | $6.5 \pm 3.8$ | $16.6 \pm 5.0$ | $7.1 \pm 2.4$ | $2.6 \pm 0.8$ | $3.9 \pm 2.4$ |
| $\boldsymbol{p}_{\mathrm{dyn}}$ in nPa | $1.7 \pm 0.6$ | $1.8 \pm 0.6$ | $1.6 \pm 0.5$ | $1.8 \pm 0.6$ | $1.7 \pm 0.8$ |
| $M_A$ | $9.4 \pm 2.4$ | $9.0 \pm 1.8$ | $9.3 \pm 1.9$ | $12.4 \pm 3.1$ | $4.9 \pm 0.9$ |
| $\beta$ | $1.7 \pm 0.8$ | $1.3 \pm 0.4$ | $1.8 \pm 0.6$ | $3.6 \pm 1.7$ | $0.4 \pm 0.2$ |

The parameters $\boldsymbol{p}_{\mathrm{dyn}}$, $M_A$ and $\beta$ are derived variables and therefore inherently connnected to the four paramters $|\boldsymbol{B}_{\mathrm{IMF}}|$, $|\boldsymbol{u}_{\mathrm{sw}}|$, $\boldsymbol{n}_{\mathrm{ion}}$, $\boldsymbol{T}_{\mathrm{ion}}$.

Transient phenomena such as interplanetary coronal mass ejections (ICMEs) are associated with the EJC type, which is typically described by high IMF magnitudes, intermediate solar wind velocities, and low Alfvén Mach numbers and plasma $\beta$s. High solar wind speeds with intermediate IMF magnitudes, Alfvén Mach numbers and plasma $\beta$ describe the CHO type solar wind (often referred to as the "fast" solar wind), which originates from open magnetic field lines in the solar corona (coronal holes). The SBO type (often referred to as the "slow" solar wind) originates from regions between the edge of coronal holes and streamer belts and can be described in terms of intermediate solar wind velocities, IMF magnitudes, Alfvén Mach numbers and plasma $\beta$. Finally, the SRR types (sometimes referred to as "very slow" solar wind) associated with the top of helmet streamers, which are cusp-like magnetic loops in the solar corona, can best be described with low solar wind velocities and IMF magnitudes and high Alfvén Mach numbers and plasma $\beta$ (see Xu and Borovsky, 2015; Borovsky, 2018b, 2020; Koller et al., 2024, and references therein). Table 1 shows the median values $\tilde{X}$ and the median absolute deviation, defined as the median of $|X_i - \tilde{X}|$, where $X_i$ are the individual data points, for different solar wind parameters extracted from our OMNI data selection, falling into the four classes to further quantify the above descriptions.

The threshold-based definition of the classes leads to problems near the class boundaries, which could lead to false classifications. To prevent these edge cases from skewing our statistics, we applied additional thresholds for each class slightly above and below the thresholds given in Xu and Borovsky (2015). We arbitrarily chose the thresholds in such a way that 5 % of each class were identified as edge cases in the classification of our entire OMNI dataset.

Since the classification scheme of Xu and Borovsky (2015) does not include information about the orientation of the IMF, which is an important factor for the response of the MP, we extended the four categories to include information about non-radial northward ( $\vartheta_{\mathrm{cone}} > 30°$ and $|\vartheta_{\mathrm{clock}}| < 90°$), non-radial southward ( $\vartheta_{\mathrm{cone}} > 30°$ and $|\vartheta_{\mathrm{clock}}| > 90°$), and quasi-radial IMF ($\vartheta_{\mathrm{cone}} < 30°$). This gives a total of 12 different solar wind categories that can be associated with MPC observations.

**Table 2.** Number of usable MPCs in the three datasets divided into separate subsets for different magnetospheric regions. The regions are divided according to the latitude and longitude angle (see text for details) into the equatorial subsolar region, the high latitude subsolar region, the equatorial flank regions and the high latitude flank regions. The table also gives a comparison between overestimated and underestimated MPCs in each dataset and subset as percentages of the original set. The expanded and overestimated MPCs are identified for two different MP models (SH98 and N22b).

| | THEMIS MPCs | | Cluster MPCs | | MMS MPCs | | Total MPCs | |
|---|---|---|---|---|---|---|---|---|
| | comp. | exp. | comp. | exp. | comp. | exp. | comp. | exp. |
| Equat. subsol. | 29,055 | | 638 | | 9,051 | | 38,744 | |
| misrepresented in SH98 | 0.9 % | 4.5 % | 2.0 %[a] | 7.5 % | 3.2 % | 6.3 % | 1.4 % | 5.0 % |
| misrepresented in N22b | 0.5 % | 8.9 % | 0.3 %[a] | 10.8 % | 1.9 % | 9.5 % | 0.8 % | 9.1% |
| High lat. subsol. | — | | 1,317 | | 729 | | 2,046 | |
| misrepresented in SH98 | — | — | 49.0 % | 0.5 %[a] | 15.8 % | 1.1 %[a] | 37.1 % | 0.7 %[a] |
| misrepresented in N22b | — | — | 5.1 % | 7.1 % | 6.9 % | 5.6 % | 5.7 % | 6.6 % |
| Equat. flanks | 62,588 | | 3,117 | | 18,082 | | 83,787 | |
| misrepresented in SH98 | 11.3 % | 2.7 % | 7.2 % | 10.5 % | 21.1 % | 2.2 % | 13.2 % | 2.9 % |
| misrepresented in N22b | 4.0 % | 7.6 % | 2.4 % | 22.0 % | 12.3 % | 6.6 % | 5.7 % | 7.9 % |
| High lat. flanks | — | | 3,347 | | 1,616 | | 4,963 | |
| misrepresented in SH98 | — | — | 42.5 % | 2.4 % | 37.8 % | — | 40.9 % | 1.6 % |
| misrepresented in N22b | — | — | 5.4 % | 12.1 % | 15.1 % | 2.2 % | 8.6 % | 8.9 % |
| Total | 91,646 | | 8,419 | | 29,478 | | 129,540 | |
| misrepresented in SH98 | 8.0 % | 3.3 % | 27.3 % | 5.5 % | 16.4 % | 3.3 % | 11.1 % | 3.4 % |
| misrepresented in N22b | 2.9 % | 8.0 % | 3.9 % | 14.9 % | 9.1 % | 7.2 % | 4.4 % | 8.3 % |

[a] These subsets have too few observations and therefore the results from them are most likely unreliable.

## 3 Results

In the following, we only consider the crossing events from the three datasets in our analysis if all relevant parameters are determined, i.e. if we have values for $\Delta r_{\perp,\mathrm{SH98}}$, $\Delta r_{\perp,\mathrm{N22b}}$, $\vartheta_{B,n}$ and a reliable classification results from the modified Xu and Borovsky (2015) scheme. This leaves us with 129,540 crossing events in a combined dataset to examine. The exact composition of this combined MPC dataset from the three missions can be found in the Tbl. 2. Note that we have not grouped multiple crossings that are close in time and space, as is often done to fit MP models. We are aware that this may lead to a bias in our distributions. However, due to our normalisation, this should not affect our results.

In addtion, we separate this large combined dataset into four distinct regions of the magnetopause over the aGSE latitude $\phi$ and longitude $\lambda$: (1) subsolar crossings observed in the region where $|\lambda| < 30°$ and $|\phi| < 30°$; (2) high latitude subsolar

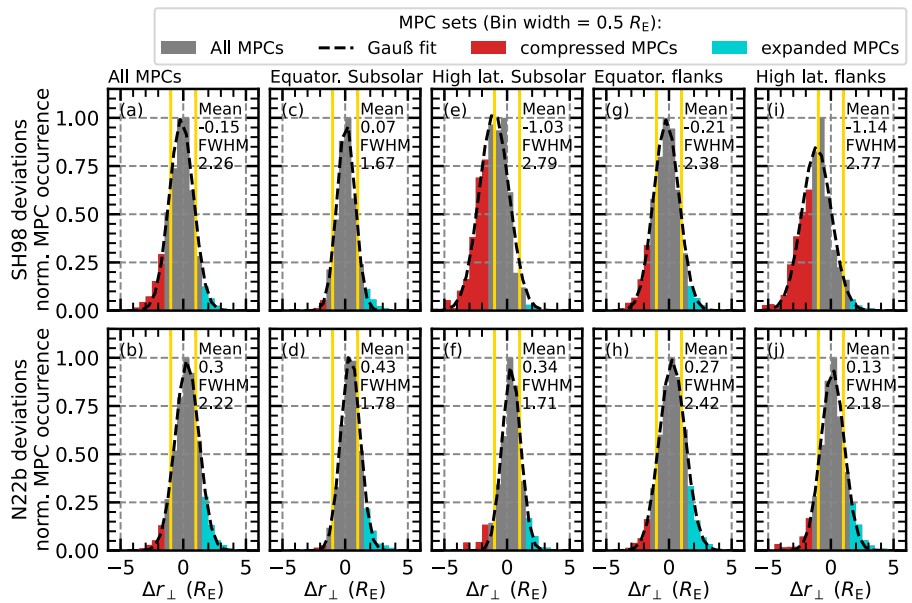

**Figure 2.** Distributions of $\Delta r_\perp$ between the observations in the combined datasets and the prediction of the SH98 model in panel (a) and the prediction of the N22b model in panel (b). The distribution from (a) and (b) are split into subsets: (c) and (d) show the distributions for the subsolar magnetopause; (e) and (f) the high latitude MP in the noon sector; (g) and (h) the flank MP observations in the equatorial plane; (i) and (j) the flank MP observations in the high latitudes. The yellow lines represent the 1 $R_E$ uncertainty of the MP models reported by Case and Wild (2013) and Staples et al. (2020). The red and cyan coloured regions of the histograms are the MPCs that clearly deviate from the selected model in the data set (see text for details). The dashed black lines represent a Gaussian fit to the histograms, with the mean/median and full width at half maximum (FWHM) of the fits also shown. The normalisation of the distribution is done first by dividing by the spacecraft dwell time in each bin and second by scaling the distribution to the maximum occurrence rate.

crossings observed in the region where $|\lambda| < 30°$ and $|\phi| \geq 30°$; (3) near-equatorial flank crossings observed in the region where $|\lambda| \geq 30°$ and $|\phi| < 30°$ holds; (4) high-latitude flank crossings observed in the region where $|\lambda| \geq 30°$ and $|\phi| \geq 30°$ holds. The total number of MPCs observed in these regions can also be found in Tbl. 2.

Based on the results of Case and Wild (2013) and Staples et al. (2020), we consider $\pm 1$ $R_E$ deviations between observed and model MP locations as typical errors of the models. Thus, deviations in $\Delta r_\perp$ up to $\pm 1$ $R_E$ are considered as part of the constant motion of the MP surface, which empirical models cannot capture, and as agreement between model and observation.

Panel (a) and (b) of Fig. 2 show the distributions of $\Delta r_\perp$, the deviation between the observation and the models being considered. In order to eliminate any orbital bias in the histogram, we used the dwell time of the spacecraft in each bin for normalisation. Here the dwell time should be seen as the time during which the spacecraft are within the different bins (i.e. between certain ranges of model deviations) and can potentially observe an MPC with the associated model deviation. The normalisation therefore provides a more realistic distribution of the occurrence of MPC deviations, as MPCs found in regions frequently visited by the spacecraft are reduced in importance.

Both distributions show a similar width according to a Gaussian fit we applied, which gives a Full Width at Half Maximum (FWHM) value of 2.26 (2.22) $R_{\mathrm{E}}$ for the SH98 (N22b) model. Thus we can identify quite a few crossings with $|\Delta r_\perp| > 1.5\,R_{\mathrm{E}}$ where the observed location differs from the predicted one in both distributions (cyan and red coloured events). For the SH98 model, 14.5 % of the events are either compressed or underestimated MPCs, and for the N22b model, 12.7 % are misrepresented MPCs. However, an important difference between the distributions is the mean/median of the fit: -0.15 $R_{\mathrm{E}}$ for the SH98 model and 0.3 $R_{\mathrm{E}}$ for the N22b model. This leads to the fact that the N22b model, when compared to the SH98 distribution, has significantly more underestimated MPCs.

Figure 2c-j show the regional separated distributions of $\Delta r_\perp$. There are a few things worth noting. In both figures it is clear that the distributions for the flank regions (panels (g)-(j)) are broader than the distributions for the two subsolar regions (panels (c)-(f)). In general, the distribution for the equatorial subsolar region with a FWHM value of 1.67 (1.78) $R_{\mathrm{E}}$, considering the SH98 (N22b) model, shows the narrowest distribution with mostly underestimated MPCs outside the error bounds of the model prediction. The distribution for the deviation from the SH98 model are shifted to negative deviations in the high latitude regions (cf. 2e and i), resulting in a lots of overestimated MPCs in these subsets. Since the SH98 model does not include a cusp indentation, the encounter of the cusp in the high latitude regions causes a bias towards overestimated MPCs in the $\Delta r_\perp$ distribution, which is clearly visible here and was reported by Grimmich et al. (2024b) for the Cluster dataset used here. The N22b model includes an indentation term for the cusp, and we can see in Fig. 2f and j that this results in a narrower distribution with drastically fewer overestimated MPCs. However, we can also see the shift towards positive deviations of $\Delta r_\perp$ in all regions for the N22b model, which of course also reduces the amount of observed overestimated MPCs in the high latitudes.

Since we calculate an estimate for $\vartheta_{B,n}$ of the bow shock for each MPC, we can show here which bow shock configuration (quasi-parallel for $\vartheta_{B,n} < 45$, quasi-perpendicular for $\vartheta_{B,n} > 45$) is favourable for the occurrence of misrepresented MPCs. We identify favourable conditions, similar to the favourable solar wind conditions identified by Grimmich et al. (2023a, 2024b) for the THEMIS and Cluster data sets. We compare the occurrence of $\vartheta_{B,n}$ associated with overestimated and underestimated MPCs with the total occurrence of different bow shock configurations over the course of the three missions.

Figure 3 again compares the result for the two different models. The distributions of $\vartheta_{B,n}$ associated with the outlying MPCs have been normalised by dividing these distributions by the reference distribution, which includes all times when a given $\vartheta_{B,n}$ value is observed and is not restricted to times when MPCs are observed. Thus, a value of one in the plots indicates that the overall distribution and that associated with the over-/underestimated MPCs are the same, and we can identify favourable conditions by looking for areas where we see values above one.

In order to be sure that the observed deviations are statistically significant and not due to chance, we performed a Mann-Whitney U test. This test is a generalisation of Student's t-test for non-normal distributions like ours and is used to analyse the differences between two independent samples with similar shapes.

The null hypothesis of the test is always that there are no significant differences between the samples, which in our case would mean that the distribution of deviant events is the same as the reference distribution. To reject or accept the null hypothesis, a rank is assigned to each observation. The ranks are then summed for each group. The test statistic U-value is calculated from these rank sums for each distribution and the smaller of the U-values is used for hypothesis testing. If the calculated U-value

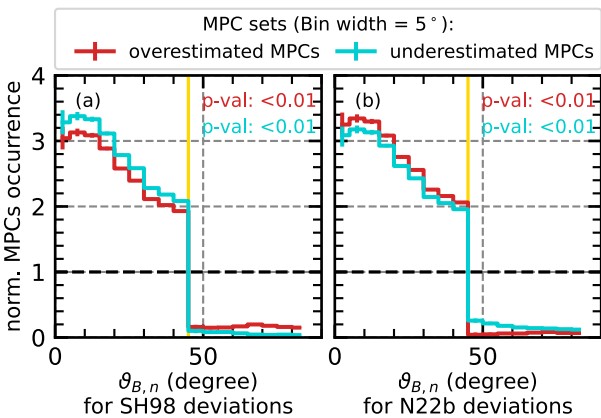

**Figure 3.** The $\vartheta_{B,n}$ distributions for the overestimated and underestimated MPCs misrepresented in the SH98 (a) and N22b (b) model are shown in red and cyan, respectively. The distributions are normalised by dividing by the total number of different $\vartheta_{B,n}$ values. A value of one in the plots (black dashed line) would therefore indicate that the occurrence of misrepresented MPCs is identical to the total occurrence, while values greater than one indicate that misrepresented MPCs occur more frequently under these conditions. The yellow lines mark the boundary between the quasi-parallel and the quasi-perpendicular foreshock condition associated with $\vartheta_{B,n}$. In addition, we show in each panel for the subsets the p-value that results from a Mann-Whitney U test: values less than 0.05, which is the case here, indicate statistically significant deviations from the reference distribution.

is less than the critical value from the Mann-Whitney distribution (based on sample sizes), the null hypothesis is rejected, indicating a significant difference between the two groups. In our case, the probability value from the test statistic must be smaller than 0.05 (see Mann and Whitney, 1947, for more details).

Here, the test results in probability values well below 0.01, thus the distribution of $\vartheta_{B,n}$ associated with deviant events is

significantly different from the overall distribution of $\vartheta_{B,n}$, which assures the seen favourable conditions for deviant events.

We can clearly see that for both models and for both the occurrence of overestimated and underestimated MPCs $\vartheta_{B,n} < 45°$ are favourable conditions. Thus, MPCs behind the quasi-parallel bow shock where foreshock has developed tend to deviate more from the MP observation. In addition, we can extract from our data set that 42 % (38 %) of the MPCs deviating from the SH98 (N22b) model predictions are associated with the quasi-parallel bow shock conditions and likely foreshock activity,

and 19 % (15 %) of the observed MPCs associated with $\vartheta_{B,n} < 45$ and $q > 0$ deviate from the SH98 (N22b) model prediction. Considering that $\vartheta_{B,n} < 60°$ is also sometimes used to define the boundary of the active foreshock region (Wilson, 2016; Karlsson et al., 2021), about 54 % of the misrepresented MPCs might be associated with foreshock activity.

Since we know that under quasi-parallel conditions the MP can be highly disturbed and the occurrence of misrepresented MPCs in both directions (overestimated and underestimated MPCs) is likely, we look again at the $\Delta r_\perp$ distribution in different

regions to see if the conditions in one region have a particular influence. Figure 4 shows, for the SH98 and N22b models respectively, the comparison of the distributions of $\Delta r_\perp$ associated with quasi-parallel and quasi-perpendicular conditions for all MPCs and in the four MP regions defined above.

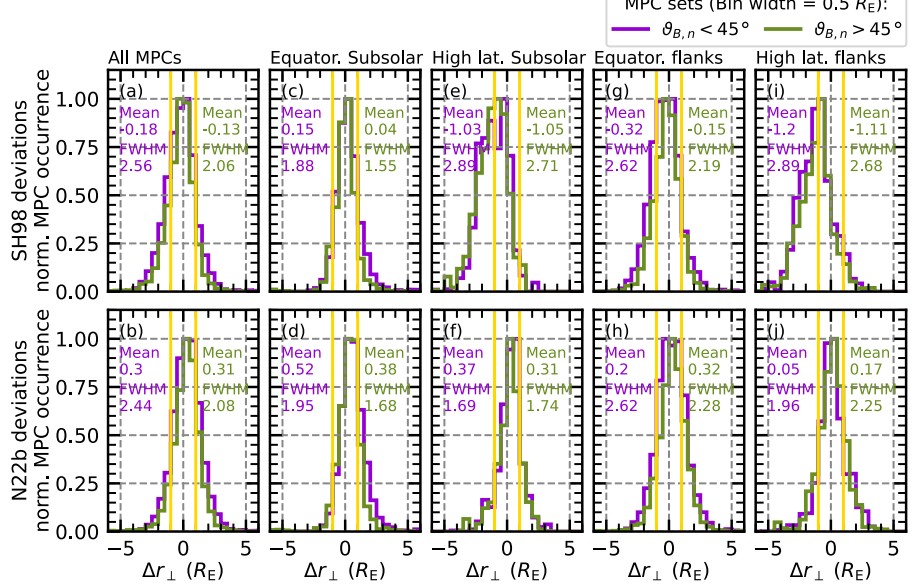

**Figure 4.** Comparison of $\Delta r_\perp$ distributions of MPCs associated with different $\vartheta_{B,n}$ in different magnetospheric regions. As before in Fig. 2, panels (a) and (b) show the distributions that include all of the MPC observations; (c) and (d) show the distributions for the subsolar magnetopause; panel (e) and(f) the high latitude MP in the noon sector; panel (g) and (h) the flank MP observation in the equatorial plane; panel (i) and(j) the flank MP observation in the high latitudes. The violet distributions belong to MPCs associated with $\vartheta_{B,n} < 45°$ and therefore observed behind a quasi-parallel foreshock region, while the green distributions belong to MPCs associated with $\vartheta_{B,n} > 45°$. For each distribution, the mean/median and full width at half maximum (FWHM) values of an associated Gaussian fit are also displayed and the yellow lines mark the reported 1 $R_E$ uncertainty of the MP model.

Figure 4a and b show a visible effect of the presence of the foreshock region on the general $\Delta r_\perp$ distributions, with negligible shifts in the mean/median but noticeable broadening of the distributions. Specifically, the bow shock condition seems to have

295 the largest effect on the subsolar region. This is somewhat to be expected, as the MP in this case is immediately downstream of the foreshock, which is not the case at higher latitudes. Both in Fig. 4c and d we notice a shift towards larger positive deviation from the model predictions if the MPCs are associated with quasi-parallel conditions; the mean/median value of the Gaussian fit for the SH98 (N22b) model distribution shifts from 0.04 (0.38) $R_E$ to 0.15 (0.52) $R_E$ when the MPCs are associated with quasi-parallel conditions. We can also see that the quasi-parallel distribution is significantly wider compared to

300 the quasi-perpendicular distribution in the sub-solar region (cf. FWHM values in panel (c)), indicating an increased variability of the MP location, e.g. due to more frequent motion.

Although we see a slight broadening of the $\Delta r_\perp$ distribution associated with low $\vartheta_{B,n}$ values for both models in the flank and high latitude regions (panels (e) to (j)), accompanied by mostly negligible shift in the distribution, the influence is not as pronounced. At the flanks the quasi-parallel distributions shift by about 0.1 to 0.15 $R_E$ towards negative deviations, while at

305 the high latitude subsolar MP the shift is about 0.02 to 0.06 $R_E$ towards positive deviations.

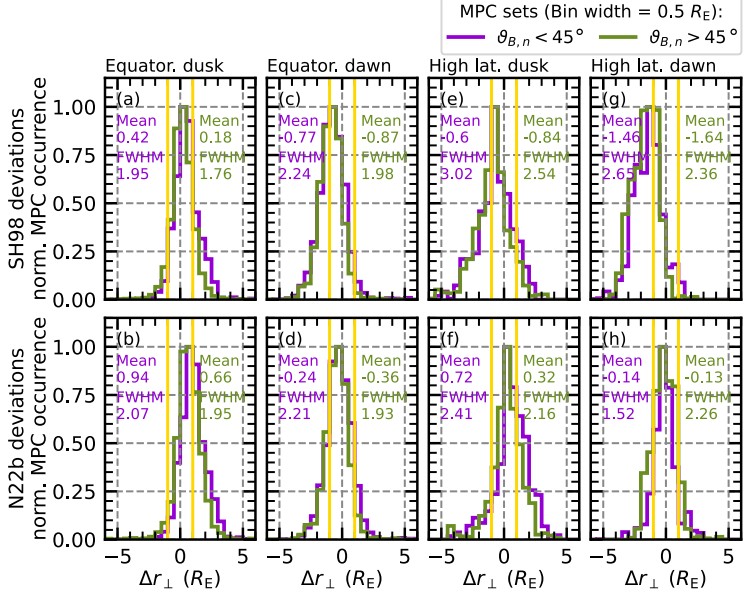

**Figure 5.** Comparison of $\Delta r_\perp$ distributions of MPCs associated with different $\vartheta_{B,n}$ on the dawn and dusk flanks. Panels (a) and (b) show the distributions for the equatorial dusk flank MPC observations; (c) and (d) show the distributions for the equatorial dawn flank; panels (e) and (f) the high latitude MP at the dusk flank; panels (g) and (h) the high latitude MP at the dawn flank. The violet distributions similar to Fig. 4 belong to MPCs associated with $\vartheta_{B,n} < 45°$, while the green distributions belong to MPCs associated with $\vartheta_{B,n} > 45°$. For each distribution, the mean/median and full width at half maximum (FWHM) values of an associated Gaussian fit are also shown, and the yellow lines mark the reported 1 $R_E$ uncertainty of the MP model.

Since general asymmetries between the dawn and dusk magnetosphere have been reported, especially with respect to the waves and undulations of the MP surface but also in internal magnetospheric properties such as plasma density(e.g. Russell et al., 1997; Nykyri, 2013; Walsh et al., 2014; Archer et al., 2015a; Henry et al., 2017), assessing both flanks in one subset might obscure different effects that are important on only one flank. We therefore decided to separate our flank subsets again into dawn and dusk. This will also give us more information on how well the two models have predicted the magnetopause at different locations, which may help to further improve one of them.

So if we look at the dawn and dusk flanks separately, we find a general asymmetry in the $\Delta r_\perp$ distributions (see Fig. 5): In the equatorial plane (panels (a)-(d)), the distributions on the dusk flank are narrower with a mean at positive deviations, whereas the distributions on the dawn flank are wider with a mean/median shifted towards negative deviations. While at high latitudes (panels (e)-(h)) the shift of the means/medians between dawn (more towards negative deviations) and dusk (more towards positive deviations) is the same as at the equatorial latitudes, the width of the distributions is different, with wider distributions at high latitude dusk flanks and narrower distributions at high latitude dawn flanks. The shifts in the means/medians indicate that, on average, the models tend to under-predict the location of the MP at dusk and over-predict the location of the MP at dawn.

In addition to these general observations on the dawn and dusk flanks, we can also see a similar widening of the distribution associated with foreshock activity (i.e. for $\vartheta_{B,n} < 45$) as in the other regions shown in Fig. 4. In the equatorial plane, the distributions on the dawn flanks widen almost a third more than the distribution on the dusk flank when foreshock activity leads to more frequent deviations from model predictions due to the more turbulent motion of the MP surface behind a quasi-parallel bow shock.

We also find something similar to the effect seen in the subsolar region, as the separation in the dawn and dusk flank crossings shows a shift of the distributions towards positive deviations (i.e. towards an underestimation of MP locations) on both flanks. This is in contrast to Fig. 4(g) and (h), where the distributions appear to be shifted towards negative deviations (i.e. towards an overestimation of MP locations), under quasi-parallel conditions. The previously observed shift towards negative values is most likely a result of the asymmetry between the two flanks, as the dawn flank crossings are generally shifted towards these

values and also have a stronger difference between quasi-parallel and quasi-perpendicular conditions.

    We would like to point out that this asymmetry could be a result of the fitting of the models used. If the models were fitted to non-aberrated coordinates, then applying them to aberrated coordinates would cause the asymmetry we see. However, to our knowledge, both models used the aberrated coordinates to fit their respective models. Therefore, application to our data should not result in the asymmetry seen. This suggests that the asymmetrie has most likely a more physical explanation like for

example the more frequent occurence of KHI and the subsequnts waves at the dawn flank MP (e.g., Kavosi and Raeder, 2015; Nykyri et al., 2017; Henry et al., 2017).

    Besides the influence of the bow shock configuration, which seems to correlate well with some of the observed deviations from model predictions, we also want to better determine the solar wind conditions responsible for the deviations. Since we have associated each MPC with a corresponding solar wind plasma class, we can investigate the occurrence of misrepresented

MPCs for a combination of solar wind parameters (instead of the single parameter influence investigated by Grimmich et al., 2023a, 2024b).

    In Fig. 6 we show the normalized occurrence of overestimated and underestimated MPCs during the different solar wind conditions based on the Xu and Borovsky (2015) scheme in the four different magnetopause regions for both models. Normalization was performed by dividing each number of occurrences of a particular class associated with an MPC by the total

345 number of occurrences of that class in the OMNI dataset between 2001 and 2024, before scaling the distribution in each panel for each model in such a way that the combined occurrence of all four types is equal to 1. While the relative abundance of classes for each panel is not affected by normalization, comparisons between panels must be made with caution, as the scaling for better visibility may distort the view on the importance of classes in different panels.

    The figure allows a direct comparison between the two models and shows mainly the agreement between the two, revealing

the most important classes of solar wind present when overestimated and underestimated MPCs occur. In order to provide further information on the solar wind conditions responsible for the occurrence of deviant MPC, the results of the separation of the four plasma classes according to the IMF direction are shown in the Tbls. 3 and 4.

    We see some clear dependencies for the occurrence of misrepresented MPCs (more or less independent of the model used to determine these MPCs): overestimated MPCs are clearly common for southward IMF orientations, with only EJC plasma

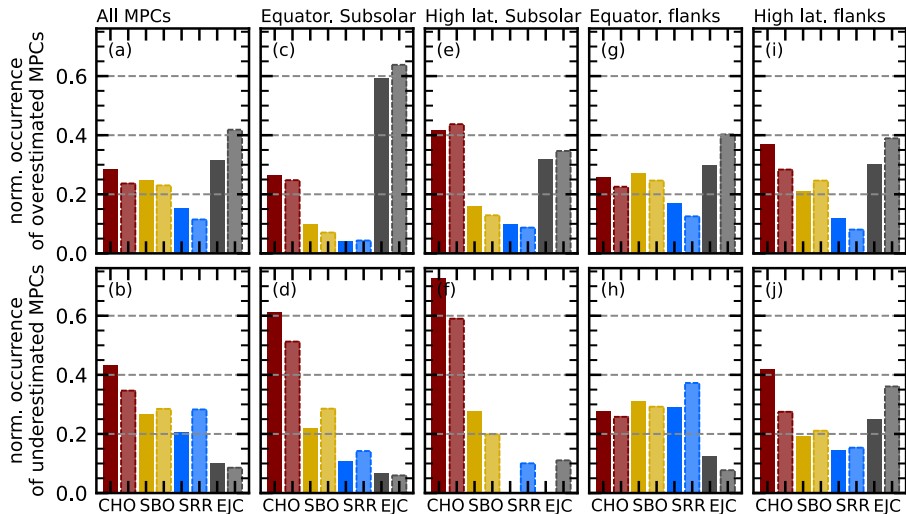

**Figure 6.** Comparison of the occurrence of overestimated MPCs (top panels) and underestimated MPCs (bottom panels) misrepresented in the SH98 and the N22b model for different solar wind plasma conditions. The distribution associated with the SH98 model is shown as solid bars, while the distributions associated with the N22b model are shown as slightly transparent bars with dashed edges. The solar wind conditions are grouped according to the classification scheme of Xu and Borovsky (2015), with different colours corresponding to different solar wind types: red for coronal hole origin (CHO), yellow for streamer belt origin (SBO), blue for sector reversal region (SRR) and grey for ejecta (EJC). Each bin is normalized by dividing the count rate of MPCs during a particular solar wind type by the count rate of that solar wind type in the OMNI data during the observation period, and then scaled for each model in such a way that the combined occurrence of all four types is equal to 1 in each panel. Panels (a) and (b) show the combined MPC datasets events, panels (c) and (d) MPC events observed in subsolar region, panels (e) and (f) events observed in high latitude subsolar region, panels (g) and (h) show events observed in equatorial flank regionsand panels (i) and (h) show events observed in high latitude flank regions.

being quite prominent. However, we also see that SBO and CHO plasma are similarly abundant when overestimated MPCs are observed (see Tbl. 3). Interestingly, the radial IMF seems to play a role especially in the high latitude regions in the CHO plasma, according to the SH98 deviations (Tbl. 3. underestimated MPCs occur most frequently under any radial IMF direction and most frequently during the CHO solar wind (see Fig. 6 and Tbl. 4). Although CHO dominates, SBO and SRR plasma become more important for underestimated MPCs on the flank (panels (h) and (j)), while EJC solar wind plays almost seemingly only for high latitude flank crossings a role (panel (j)).

It should be noted that there are obvious differences between the results associated with the SH98 and N22b models (e.g. in panels (i) and (j)). However, these differences can easily be attributed to the lack of data for overestimated and underestimated MPCs associated with one or the other model in certain magnetospheric regions (see Tbl. 2).

Similar to the $\Delta r_\perp$ distributions, we want to separate the distribution of the occurrence of misrepresented MPCs associated with different solar wind classes into MPCs behind a quasi-parallel and a quasi-perpendicular bow shock using $\vartheta_{B,n}$. Therefore, Fig. 7 shows the occurrence under the solar wind classes for MPCs with $\vartheta_{B,n} < 45°$ and $\vartheta_{B,n} > 45°$; the overestimated and

**Table 3.** Comparison of the relative occurrence rates of overestimated MPCs deviating from the SH98 and N22b models for different solar wind plasma conditions. The solar wind conditions are grouped according to the classification scheme of Xu and Borovsky (2015): coronal hole origin (CHO), streamer belt origin (SBO), sector reversal region (SRR) and ejecta (EJC). Each solar wind type is further divided into three subcategories corresponding to the IMF direction: quasi-radial IMF, northward IMF and southward IMF. The maximum occurrence rates for a given direction in each row are highlighted in bold. The values for the general solar wind classes are taken from Fig. 6, while the IMF direction indicates which of the directions is the most common relative to the occurrence of the type.

| comp. MPCs (SH98 / N22b) | Equat. subsol. | High lat. subsol. | Equat. flanks | High lat. flanks | All Regions |
|---|---|---|---|---|---|
| CHO with | 26.64 % / 24.78 % | **41.87 % / 43.73 %** | 25.76 % / 22.56 % | **36.91 %** / 28.37 % | 28.39 % / 23.66 % |
| rad. | **70.23 % / 71.66 %** | **42.83 %** / 37.21 % | **38.92 %** / 34.05 % | **34.6 %** / 33.12 % | **40.48 %** / 38.04% |
| north. | 6.92 % / 5.31 % | 26.57 % / 11.65 % | 28.09 % / 26.48 % | 31.34 % / 28.27 % | 27.19 % / 23.79 % |
| south. | 22.84 % / 23.03 % | 30.6 % / **51.14 %** | 32.99 % / **39.47 %** | 34.06 % / **38.61 %** | 32.33 % / **38.17 %** |
| SBO with | 9.79 % / 7.05 % | 16.16 % / 12.89 % | 27.21 % / 24.65 % | 20.98 % / 24.63 % | 24.7 % / 23.01 % |
| rad. | **62.17 % / 66.41 %** | **37.38 %** / 0.0 % | 25.59 % / 23.9 % | 28.11 % / 12.83 % | 27.48 % / 24.31 % |
| north. | 15.23 % / 14.26 % | 26.13 % / 24.53 % | 35.96 % / 37.58 % | 30.88 % / 19.7 % | 34.3 % / 35.3 % |
| south. | 22.6 % / 19.33 % | 36.49 % / **75.47 %** | **38.45 %** / 38.52 % | 41.02 % / **67.47 %** | **38.22 %** / 40.39 % |
| SRR with | 4.13 % / 4.35 % | 10.04 % / 8.72 % | 17.14 % / 12.53 % | 11.8 % / 8.05 % | 15.23 % / 11.47 % |
| rad. | 24.27 % / 0.0 % | **39.52 %** / 0.0 % | 29.88 % / 30.5 % | 33.29 % / 22.24 % | 30.61 % / 28.98 % |
| north. | **60.8 % / 70.37 %** | 28.06 % / **50.45 %** | 32.64 % / 33.33 % | 29.31 % / 13.93 % | 32.4 % / 33.45 % |
| south. | 14.93 % / 29.63 % | 32.42 % / 49.55 % | **37.48 %** / 36.17 % | 37.4 % / **63.84 %** | **36.94 % / 37.57 %** |
| EJC with | **59.43 % / 63.81 %** | 31.93 % / 34.66 % | 29.89 % / **40.26 %** | 30.31 % / **38.95 %** | 31.96 % / **41.86 %** |
| rad. | 8.48 % / 0.0 % | 24.7 % / 0.0 % | 37.29 % / 16.47 % | **43.25 %** / 33.92 % | 35.38 % / 16.03 % |
| north. | 25.85 % / 2.44 % | **39.02 %** / 4.99 % | 22.1 % / 20.35 % | 28.59 % / 22.97 % | 24.28 % / 18.5 % |
| south. | **65.66 % / 97.56 %** | 36.28 % / **95.01 %** | **40.61 % / 63.18 %** | 28.16%/ **43.11 %** | **40.34 % / 65.46 %** |

underestimated MPCs are selected with the SH98 and N22b models, respectively. For both models we can again see similar characteristics for the occurrence of misrepresented MPCs.

We can see more clearly that for the overestimated MPCs (top panels of Fig. 7) the southward IMF and EJC plasma is responsible for deviations behind quasi-perpendicular bow shock conditions, while radial IMF conditions within all plasma types occur often for the events behind the quasi-parallel bow shock. Furthermore, CHO plasma appears to be more important

**Table 4.** Comparison of the relative occurrence rates of underestimated MPCs deviating from the SH98 and N22b models for different solar wind plasma conditions. The solar wind conditions are grouped according to the classification scheme of Xu and Borovsky (2015): coronal hole origin (CHO), streamer belt origin (SBO), sector reversal region (SRR) and ejecta (EJC). Each solar wind type is further divided into three subcategories corresponding to the IMF direction: quasi-radial IMF, northward IMF and southward IMF. The maximum occurrence rates for a given direction in each row are highlighted in bold. The values for the general solar wind classes are taken from Fig. 6, while the IMF direction indicates which of the directions is the most common relative to the occurrence of the type.

| exp. MPCs (SH98 / N22b) | Equat. subsol. | High lat. subsol. | Equat. flanks | High lat. flanks | All Regions |
|---|---|---|---|---|---|
| CHO with | **61.19 % / 51.26 %** | **72.49 % / 59.01 %** | 27.7 % / 25.8 % | **41.87 %** / 27.49 % | **43.04 % / 34.66 %** |
| rad. | **47.04 % / 52.29 %** | **63.48 % / 59.96 %** | 62.35 % / 55.54 % | **55.6 % / 55.47 %** | **52.96 % / 54.1 %** |
| north. | 15.75 % / 15.98 % | 7.45 % / 10.68 % | 21.74 % / 28.31 % | 16.61 % / 15.5 % | 17.93 % / 21.51 % |
| south. | 37.21 % / 31.73 % | 29.08 % / 29.36 % | 15.91 % / 16.15 % | 27.79 % / 29.04 % | 29.11 % / 24.38 % |
| SBO with | 21.77 % / 28.55 % | 27.51 % / 19.87 % | **30.84 %** / 29.22 % | 19.07 % / 21.09 % | 26.56 % / 28.48 % |
| rad. | **75.75 % / 70.24 %** | 35.99 % / **46.51 %** | **67.17 % / 60.57 %** | **66.44 % / 54.05 %** | **70.42 % / 63.65 %** |
| north. | 10.25 % / 15.06 % | 0.0 % / 25.36 % | 17.18 % / 22.63 % | 21.74 % / 18.36 % | 14.5 % / 19.85 % |
| south. | 14.0 % / 14.7 % | **64.01 %** / 28.13 % | 15.65 % / 16.8 % | 11.82 % / 27.59 % | 15.08 % / 16.5 % |
| SRR with | 10.57 % / 14.22 % | 0.0 % / 10.06 % | 28.95 % / **37.26 %** | 14.29 % / 15.35 % | 20.38 % / 28.29 % |
| rad. | **72.43 % / 70.03 %** | — / 0.0 % | **66.69 % / 56.3 %** | **64.34 % / 67.83 %** | **68.1 % / 59.19 %** |
| north. | 19.24 % / 18.97 % | — / 100 % | 12.6 % / 21.21 % | 26.86 % / 20.91 % | 14.45 % / 20.97 % |
| south. | 8.33 % / 11.0 % | — / 0.0 % | 20.71 % / 22.49 % | 8.8 % / 11.26 % | 17.46 % / 19.84 % |
| EJC with | 6.47 % / 5.97 % | 0.0 % / 11.07 % | 12.52 % / 7.73 % | 24.78 % / **36.07 %** | 10.02 % / 8.57 % |
| rad. | 0.0 % / 19.32 % | — / 0.0 % | 0.0 % / 36.03 % | **74.71 % / 54.16 %** | 9.7 % / 37.28 % |
| north. | **72.89 % / 55.58 %** | — / 49.97 % | 70.24 % / **40.66 %** | 25.29 % / 38.96 % | **65.11 % / 43.07 %** |
| south. | 27.11 % / 25.1 % | — / **50.03 %** | 29.76 % / 23.31 % | 0.0 % / 6.88 % | 25.2 % / 19.65 % |

for the quasi-parallel conditions than for the quasi-perpendicular conditions. It also seems that SBO and CHO plasma types are similarly abundant only for overestimated MPCs behind the quasi-perpendicular bow shock (see panels (e) and (g)).

For the underestimated MPCs (bottom panels of Fig. 7) we see the importance of the CHO plasma for the occurrence independent of the bow shock conditions. We also see more clearly that radial IMF conditions could be mainly responsible for the underestimated MPCs behind the quasi-parallel bow shock (see panels (b) and (d)). Overall, we can see that in 50 % (49 %)

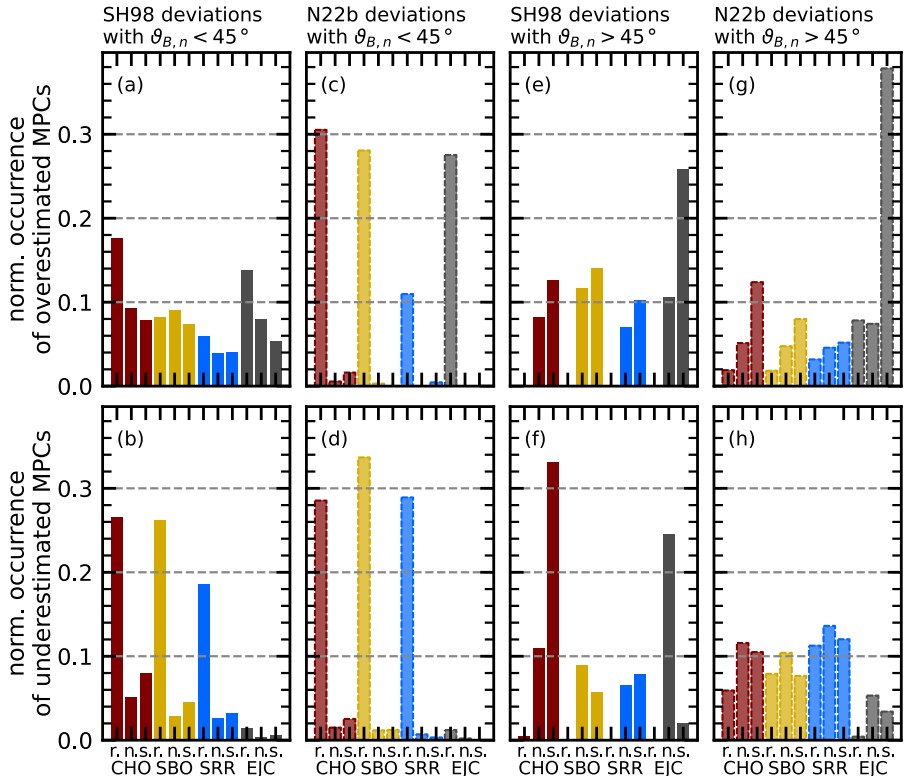

**Figure 7.** Comparison of the occurrence of overestimated MPCs (top panels) and underestimated MPCs (bottom panels) misrepresented in the SH98 and N22b MP models for different solar wind plasma conditions, similar to Fig. 6Here the solar wind classes are also separated for the IMF orientation with quasi-radial IMF (r.), northward IMF (n.) and southward IMF (s.). The occurrence of misrepresented MPCs associated with $\vartheta_{B,n} < 45°$ (panels (a) and (b) for SH98 deviations and (e) and (f) for N22b deviations) is compared to the occurrence of misrepresented MPCs associated with $\vartheta_{B,n} > 45°$ (panels (c) and (d) for SH98 deviations and (g) and (h) for N22b deviations).

of the cases we encounter underestimated MPCs that deviate from the SH98 (N22b) model behind a quasi-parallel bow shock, the IMF orientation is quasi-radial. Otherwise, it is interesting to note that southward IMF conditions (mostly in CHO plasma) seem to be quite common for underestimated MPCS behind the quasi-perpendicular bow shock for the SH98 deviations (see panel (f)). According to panel (h), the SRR plasma and the northward IMF occur simultaneously with the deviation from the N22b model predictions, in stark contrast to the SH98 results. This may be an effect of the general underestimation of MP location in the N22b model, resulting in more underestimated MPCs deviating from this model compared to the SH98 model.

## 4    Discussion

Our investigation aimed to better identify the reasons why the spacecraft-observed position of the MP surface can differ quite substantially from the empirical model predictions. We build on the results of Grimmich et al. (2023a, 2024b) and combine

three large datasets of dayside MPCs, including data from the Cluster, THEMIS and MMS missions, to comprehensively examine the dayside MP over a wide range of longitudes and latitudes. However, the different sizes of the data sets and the orbital inclinations of the missions have the disadvantage that not all magnetospheric regions are covered equally, as can be seen in Tbl. 2, which might affect occurrence rates. Nevertheless, the different orbits are also beneficial, as the bias of uneven coverage of annual solar wind conditions due to variations in spacecraft apogees causing misinterpretation of occurrence rates, as reported by Vuorinen et al. (2023), should be substantially reduced by combining the different data sets. Thus, our results from the combined dataset should overall be better at revealing the influences of the solar wind on the position of the MP than previous studies, using only a single mission dataset.

However, another orbital bias of concern could be due to the apogee of the spacecraft being lower than the average position of the boundary. As a result, spacecraft can only sample the MP at certain distances, which may belong to the innermost transient excursions or intervals of high solar wind pressure (e.g., Němeček et al., 2020). This is especially a problem in the flanks near the terminator, which has a nominal MP location of about 14.5 $R_\mathrm{E}$, while the spacecraft of e.g. THEMIS have apogees of about 13.2-13.7 $R_\mathrm{E}$. This can obviously lead to more frequent observations of overestimated MPCs and very few observations of underestimated MPCs at the flanks, and explain the shift in the distribution towards negative deviations from the MP models. Furthermore, this bias could also be a factor in the difference between the dawn and dusk flanks that we highlighted in Fig. 5. Therefore, our results regarding the magnetic flanks must be treated with caution and should be further investigated in the future.

Another point that helps to generalise and understand the occurrence of misrepresented MPCs is that we have chosen to use two different empirical MP models to identify deviant boundary crossing events. The Shue et al. (1997, 1998) model is one of the simpler models and, despite its deficiencies, is widely used in the community (e.g., also in the previous studies of Grimmich et al., 2023a, 2024b) to represent the basic behaviour of the MP surface and therefore was included here. The Nguyen et al. (2022a, b) model claims to be one of the most accurate empirical models, including all kinds of observed asymmetries and also the important cusp indentations in the high-latitude regions for MP surface modelling. Given that the cusp indentation term, adapted from Liu et al. (2015), used in the N22b model is more accurate than the one used in the Lin et al. (2010) model (Nguyen et al., 2022a), we have chosen the N22b model over the Lin et al. (2010) model for the comparison in this study. To our knowledge this model has not been validated on independent data, which is another reason why we have decided to include it here to see if it is indeed more accurate. Other models, such as the Petrinec and Russell (1996) model, were not considered here because they focus on the nightside, whereas our study is limited to the dayside.

The comparison of the two models has allowed us to see whether the occurrence of misrepresented MPCs is model dependent or whether there is a lack of fundamental physical understanding in the models. Despite the fact that the two MP models used were developed on very different data sources and have different input parameters, the occurrence of model deviations and the associated conditions that may be responsible are very similar. Indeed, this seems to indicate systematic biases due to uncaptured physics in the models, although since the MP is almost always in motion, some scatter is to be expected and will remain even if the models can be improved by our results.

We also want to address the problem that we are ignoring the time history of the solar wind. There are processes, such as the erosion of the MP towards a quasi-stationary endpoint closer to Earth, that occur over a longer time frame rather than instantaneously as implemented in the average models used. Some of our deviant events may be due to such an effect, as the solar wind used for the modelling already suggests a different location towards which the observed MP is actually still moving.

    As shown in Fig. 2a and b, the distribution of the model deviation $\Delta r_\perp$ has an almost identical width of 2.26 $R_E$ and
2.22 $R_E$, and for both models 12 to 14 % of the crossings are misrepresented MPCs, indicating that their general occurrence seems to be model independent. Since several studies (e.g., Šafránková et al., 2002; Case and Wild, 2013; Nguyen et al., 2022b; Aghabozorgi Nafchi et al., 2024) point out similar uncertainties inherent in the empirical modelling (in part due to the constant motion around an average location of the MP), this is not surprising. The obvious difference in the distribution of misrepresented MPCs between the two models (with the SH98 model we get more overestimated MPCs, while the N22b model
identifies more underestimated MPCs; see Tbl. 2) stems probably from two sources (in addition to the aforementioned orbital biases): 1) The cusp encounters in the high latitude crossings observed by Cluster lead to a bias towards overestimated MPCs for the non-indented SH98 model (see e.g., Boardsen et al., 2000; Šafránková et al., 2002, 2005; Grimmich et al., 2024b). 2) In general, the N22b model seems to under estimate the location of the MP surface for about 0.3 $R_E$ and therefore has a clear bias towards extended MPCs.

There could be a number of reasons for this. One possibility is that the functional form, and in particular the dependence of the IMF on the stand-off distance, is not appropriately chosen in the N22b model. As mentioned above, the effect of the IMF $B_z$ component on the stand-off distance is visibly weaker compared to that of the SH98 model. This results in a much less pronounced expansion of the dayside MP for the northward IMF and a weaker MP erosion for the strong southward IMF. As a result, the comparative distribution between observed and modelled MP locations for the N22b model may be visibly
shifted relative to the SH98 model distributions as seen in our plots. Another reason why the N22b model underestimates the MP location could be that the fitting of this model was influenced by the orbital apogee bias. If only the innermost crossings further away from the average MP location were included in the fitting procedure, this would result in a model trained to always predict distances closer to Earth.

    Overall, the SH98 model (despite the obvious bias from the cusp indentations and the problem of apogee coverage) is better
at predicting average MP location than the N22b model. Nevertheless, the effects on the misrepresented MPCs are visible in both models, and in the following we will only point out and discuss what is visible for both models.

    Although, for example, Aghabozorgi Nafchi et al. (2024) suggests that the deviations between observations and models are primarily due to inaccurate propagation of solar wind parameters from the L1 point (measurement point of the OMNI dataset) to Earth, we may have found other but also adjacent possible explanations in this study. As Figs. 3, 4 and 5 show, MPCs
associated with quasi-parallel bow shock conditions (i.e. $\vartheta_{B,n} < 45$) are quite often deviant crossing events. This suggests that the development of the foreshock region is an important factor in the occurrence of these events. The foreshock region strongly modifies the upstream solar wind conditions affecting the magnetosphere due to its turbulence, but also due to the occurrence of unpredictable transients (Fairfield et al., 1990; Russell et al., 1997; Plaschke et al., 2013; Walsh et al., 2019;

Zhang et al., 2022). This nicely explains the discrepancies, as due to the foreshock modifications, the input to the models are
not the conditions in the vicinity of the MP that determine the boundary position.

Here we propose that about 40-60 % (depending on the limit of $\vartheta_{B,n}$ chosen for the foreshock activity) of the deviant cases in the combined dataset are associated with foreshock activity and therefore likely to be explained by the foreshock influence. This influence seems to be strongest for the subsolar MP, and we can estimate from comparing the means in Fig. 4c and d that the MP is on average 0.12 $R_{\mathrm{E}}$ more sunward for these MPCs associated with foreshock activity. In addition, for the equatorial flanks (Fig. 5(a)-(d)) we see an asymmetry between dawn and dusk: the MP under foreshock influences is on average about 0.26 $R_{\mathrm{E}}$ more anti-Earthward on the dusk flank and on average about 0.11 $R_{\mathrm{E}}$ more anti-Earthward on the dawn flank. Furthermore, the MP clearly shows more motion on the dawn flank under foreshock influence, as the distribution of model deviations is wider compared to the dusk flank.

However, not all MPCs associated with the quasi-parallel bow shock conditions are affected by the foreshock in an extreme way, only 17 % of the crossings behind a foreshock can be identified as misrepresented MPCs, while in the other case the model predictions seem to agree with the observation and the foreshock might cause only a smaller amplitude motion around the mean location. Therefore, the presence of the foreshock does not guarantee the occurrence of deviant MPC.

It is also important to note again that our $\vartheta_{B,n}$ estimate may not be the angle at the bow shock, which is upstream of the MP undulation. The shocked solar wind plasma propagates through the magnetosheath along streamlines (e.g., Russell et al., 1983; Luhmann et al., 1986), and depending on the orientation of the IMF, this may lead to regions of the MP affected by the quasi-parallel shock, although our estimate indicates a quasi-perpendicular shock (or vice versa). We tried to validate our estimates of the foreshock state using a second method by Petrinec et al. (2022) which largely agrees with our initial guesses. Thus, the effect of a misclassified upstream bow shock configuration may not be severe.

In addtion, Kavosi and Raeder (2015) have shown in a statistical study that KHI forms more or less independently of the IMF cone angle, and therefore most likely also from the foreshock activity. Since large KHI waves can be responsible for the observation of misrepresented MPCs (as, e.g., discussed in Grimmich et al., 2023a), the misrepresented MPCs most likely to be influenced by the foreshock may, to some extent, actually be caused by KHI. However, it is still unknown to what extent the convected foreshock oscillation could influence the development of the KHI. Thus, it is not easy to determine which of these two is the source of the deviating MPC observations and future studies should aim to separate this effect and possibly assess to what extent the foreshock triggers the KHI and to what extent the KHI occurs independently.

By looking at the influences of the solar wind classes introduced by Xu and Borovsky (2015) on the occurrence of deviating MPCs, we can potentially identify and differentiate additional sources of deviations from the models. In Figs 6 and 7 we can see which solar wind parameter combinations are most likely to be present during the encounter with misrepresented MPCs.

A CHO plasma, described by high solar wind speeds and temperatures with low ion densities and intermediate IMF magnitudes (compared to the average parameter values from the OMNI data), is most often present when underestimated MPCs occur. Consistent with this finding, high solar wind speeds (independent from other parameters) have previously been reported to be favourable for the onset of (large) sunward MP deformation (Grimmich et al., 2023a, 2024b; Guo et al., 2024). This result now allows us to estimate the magnitude of other solar wind parameters that typically occur at high solar wind speeds as well.

Another point often found in the literature is the expansion of the MP under quasi-radial IMF conditions (Merka et al., 2003; Suvorova et al., 2010; Samsonov et al., 2012; Park et al., 2016; Grygorov et al., 2017; Guo et al., 2024), and so this orientation is naturally often associated with underestimated MPCs, which is also visible in our findings. In addition, our results show that the underestimated MPCs that occur during quasi-radial IMF conditions are significantly more often associated with quasi-parallel bow shock conditions. Since, under radial IMF conditions, the foreshock develops upstream of the subsolar bow shock to where the IMF is tangent to the bow shock surface, and thus most of the dayside magnetosphere would be behind a quasi-parallel bow shock, this observation is not surprising.

In combination with the likely presence of CHO plasma simultaneously with quasi-radial IMF our findings further emphasizes that, besides the "normal" turbulence influencing the MP, foreshock transients might often be responsible for the misrepresented MPCs. These transients occur more frequently in the foreshock under exactly these conditions (Chu et al., 2017; Vu et al., 2022; Zhang et al., 2022; Xirogiannopoulou et al., 2024) and several studies have already shown that the very different plasma parameters in the core of these transients can significantly deform the MP towards the Sun (e.g., Sibeck et al., 1999; Turner et al., 2011; Archer et al., 2015b; Grimmich et al., 2024c). It is also worth noting that the foreshock developed in the CHO plasma should be further investigated. This could explain exactly why the MP appears to move globally outwards under radial IMF, which is most likely accompanied by CHO plasma. The composition of this plasma group could therefore be the dominant factor missing in the explanation. However, a recent study by Lee et al. (2024) showed that for "fast" solar wind the IMF becomes more often quasi-radial, and discontinuities in the solar wind are more often oriented in a direction where they can produce larger foreshock transients when hitting the Earth's bow shock. Thus, this favourable orientation and the growth of the transients under the quasi-radial IMF conditions may be the reason that allows us to observe the misrepresented magnetopause locations caused by the transinets.

Contrary to the finding of Grimmich et al. (2023a), which suggests that high Alfvén Mach numbers and solar wind plasma $\beta$ are also important for the occurrence of underestimated MPCs, we find that the SRR plasma described by these conditions is less important and actually only relevant together with the radial IMF, which is likely to be the dominant effect for the occurrence of misrepresented MPCs. This shows that our classification of the solar wind and looking at the influences from combined datasets helps to distinguish the more important mechanisms.

For the underestimated MPCs associated with quasi-perpendicular bow shock conditions, and therefore probably not caused by the foreshock modification, we see that for the SH98 model deviations, southward CHO plasma is mostly present during the observations. Since we see that this occurrence of southward CHO plasma is associated with the subsolar region at low and high latitudes (Tbl. 4), one explanation for the underestimated MPCs could be that large flux transfer events (Elphic, 1995; Dorelli and Bhattacharjee, 2009; Fear et al., 2017) resulting from ongoing reconnection at the MP nose under southward IMF lead to displacements of the MP surface. However, these results are largely due to the high latitude underestimated MPCs, which are rather sparse in our dataset compared to the other regions. Therefore, there is no guarantee that our findings would hold up with more data points in these regions.

Interestingly, the results obtained from the N22b model deviations for the underestimated MPCs in quasi-perpendicular configuration show a rather different picture. Here, the SRR plasma occurs in virtually every IMF orientation most of the time,

although the north- and southward CHO and SBO plasmas are also very similar in abundance to the SRR plasma. It is likely that most of these deviant crossings are observed in the equatorial flank regions, as we see that this is where the SRR plasma is often seen alongside deviant events form the N22b model (see Tbl. 4). Since there are no clear favourable conditions for the occurrence of misrepresented MPCs, they appear to occur almost independently of solar wind conditions, which may point to universal effects such as KHI as the source of expansion for these events. Further research is needed to determine whether the results of the SH98 or N22b model are more reliable. However, as we observed more of the underestimated MPCs from the N22b model compared to the SH98 model due to the prediction bias of the N22b model, these results may not be as reliable as those from the SH98 model.

Another expected observation is the frequent presence of EJC plasma and southward IMF orientations during overestimated MPCs, especially for events behind a quasi-perpendicular bow shock (Fig. 7e and f). EJC plasma described (compared to the average parameter values from the OMNI data) by high IMF magnitudes and intermediate solar wind velocities, densities and temperatures is associated with strong transient phenomena like ICMEs. Such ICMEs are known to cause geomagnetic storms (e.g., Denton et al., 2006; Kilpua et al., 2017) in which the MP moves towards Earth. Similarly, it is well known that reconnection occurs during the southward IMF, leading to an MP found further Earthward (Levy et al., 1964; Paschmann et al., 1979; Sibeck et al., 1991; Shue et al., 1997, 1998; Paschmann et al., 2013). We think it is likely that overestimated MPCs may be produced in both instances. Either the reconnection fluxes are not correctly represented in the model (e.g. as in the N22b model), or the transient nature of the EJC plasma strongly disturbs the MP surface in ways that the models cannot predict, resulting in the observation of overestimated MPCs. Since we can infer that the occurrence of EJC plasma is slightly more likely than other types for southward IMF, the EJC plasma characteristic, as shown in Tbl. 1, may also be a factor in explaining the overestimated MPCs.

EJC plasma is also associated with low Alfvén Mach numbers and plasma $\beta$ caused by the high IMF magnitudes. Thus the results here showing that this type of plasma is favoured for the occurrence of overestimated MPCs agree with the previous results from Grimmich et al. (2023a) claiming exactly this from the single parameter study. However, it is now clearer that the parameters are in fact related and, combined, are responsible for the occurrence. It should be noted that the classification of EJC plasma may also be biased, as this type is very sensitive to which phase of an ICME is collected in the data. Therefore, more research should be done to validate these results.

Looking at the overestimated MPC associated with the foreshock activity, we see that in addition to the presence of the EJC plasma as the source, the radial IMF in the CHO plasma is important. This similar to the result for the underestimated MPCs is most likely linked to the foreshock appearing in front of the MP nose and foreshock transient modulating the MP. In particular, the boundary compression regions of the transients (Schwartz, 1995; Turner et al., 2013; Liu et al., 2016) cause MP motion towards the Earth, which may result in overestimated MPC observations.

In addition, the quasi-parallel domain of the bow shock is cited as the origin for the development of magnetosheath jets (Plaschke et al., 2018). Since they can lead to MP indentation even under radial IMF conditions where the MP is expected to be more expanded (e.g. Shue et al., 2009; Wang et al., 2023; Yang et al., 2024; Němeček et al., 2023), they could be another explanation for the occurrence of overestimated MPCs.

In general, we can also see that SBO plasma (the "normal"/"mean" solar wind) is often present during overestimated MPC observations. Processes such as Kelvin-Helmholtz or surface waves, which occur independent from IMF orientations (Johnson et al., 2014; Kavosi and Raeder, 2015; Masson and Nykyri, 2018; Archer et al., 2019, 2024a), are possible explanations for these events, especially since most of the overestimated MPCs more often associated with SBO plasma are observed on the flanks where KHIs are more likely.

Overall, the SW class analysis gives a very similar picture to the results of Grimmich et al. (2023a). However, it now seems clearer to what extent the foreshock, which is often overlooked when discussing uncertainties in MP modelling, should be held responsible and in which region this phenomenon might be important for the occurrence of misrepresented MPCs.

Nevertheless, we would like to point out again that some regional results could be biased due to the few events observed, especially the high latitude regions would benefit from more events to further solidify the results obtained. The upcoming Solar Wind Magnetosphere Ionosphere Link Explorer (SMILE) mission (Branduardi-Raymont et al., 2018) will be one of the next big magentospheric missions with a highly inclined (70°) and elliptical ($1 \times 20\ R_{\mathrm{E}}$) orbit around the Earth. Thus, this mission may be able to provide new in-situ observations of the high latitude MP in this region that could be used to reduce this potential bias.

Furthermore, the SMILE mission aims to observe and image the MP via X-ray observations and is in need of accurate MP models for its analysis techniques (see Kuntz, 2019; Wang and Sun, 2022). Our study can be seen as a first step towards developing a better empirical model that captures to some extent the effect presented here. An important point for such a future model could be a regional dependency, as we have seen that deviations are more common on the flanks, and the inclusion of foreshock activity by including $\vartheta_{B,n}$. In addition, a more probabilistic approach to prediction of the MP surface under different input parameters may be beneficial.

## 5 Conclusions

To sum up, by combining data from three different spacecraft missions that have collected MP observations over the last two decades from 2000 to 2024, including two full solar cycles,, we have been able to identify model-independent conditions during deviations between model predictions and spacecraft observations. The model deviations are present throughout the dayside magnetosphere, although regional dependencies are clearly visible. In the magnetospheric flanks the deviations are generally more frequent, especially the overestimated MPCs, while the underestimated MPCs seem to occur more frequently in the near equatorial plane. However, it is likely that this observation is due to observations from spacecraft orbits with limited apogee not being able to properly sample the flank magnetopause.

In general, our statistics show that foreshock/quasi-parallel shock conditions are conducive to misrepresented MPCs, even if they are not directly caused by the foreshock itself, with the most pronounced effect in the subsolar region. The turbulent nature of the foreshock and the occurring transients may lead to large displacements of the MP in earthward and anti-earthward directions, generally resulting in an average model deviation of 0.1 to 0.2 $R_{\mathrm{E}}$ anti-earthward. This also leads us to suspect that large amplitude surface and Kelvin-Helmholtz waves may be more common, and our results may often represent the resulting

moving MP boundary. In fact, it is possible that in those cases where we think the foreshock is responsible for the deviant MPC observations, KHI, which developed independently of the foreshock, is the real cause. Thus, more research is needed to further define the complex interactions in the magnetospheric system and to improve our understanding of the foreshock effect on MP
motion.

For example, in future studies we would like to examine surface waves in relation to the foreshock to see how the amplitude of the MP motion might change and whether this can explain some of the misrepresented MPCs. This may be similar to studies such as Song et al. (1988) or Russell et al. (1997), which have already examined the amplitude of MP motion in relation to the foreshock in some way. However, the dataset used here also gives us the opportunity to examine the changing behaviour
over several solar cycles, which may reveal interesting behaviour. In addition, it is clearly necessary to determine whether the development of KHI is modified by the foreshock or whether it actually occurs independently of the foreshock, and thus may more often be the actual cause of the deviating MPCs.

Confirming and updating the results of Grimmich et al. (2023a), we further propose that overestimated MPCs may favourably occur during southward IMFs embedded in a plasma of high IMF magnitudes caused by solar transients such as ICMEs, when
foreshock activity is not a reasonable cause; overestimated MPCs may occur due to foreshock activity specifically for "fast" solar wind with radial IMF orientation; underestimated MPCs generally occur most frequently for the "fast" solar wind, with foreshock activity responsible for deviations under radial IMF.

Overall, this study has identified processes that are still missing from commonly used MP models, and may help to improve these models in the future. However, as some of these identified processes may be associated with transient phenomena in the
foreshock, which are inherently difficult to predict, this will be a challenging endeavour.

*Data availability.*  The Open Science Framework (OSF) hosts the assembled MPC dataset by Grimmich et al. (2024a) for Cluster C1 and C3 at https://osf.io/pxctg/ and the dataset by Grimmich et al. (2023b) for THEMIS at https://osf.io/b6kux/. The dataset by Toy-Edens et al. (2024a) is available on Zenodo following https://zenodo.org/records/10491878. The OMNI data (King and Papitashvili, 2005) were obtained from the GSFC/SPDF OMNIWeb interface at https://spdf.gsfc.nasa.gov/pub/data/omni/omni_cdaweb/. The Dst index (Nose et al., 2015)
used in this paper was provided by the WDC for Geomagnetism, Kyoto http://wdc.kugi.kyoto-u.ac.jp/wdc/Sec3.html

*Author contributions.*  NG performed the analysis and wrote the original manuscript with additional input from AP and MOA. DGS together with FP was involved in developing the research idea for this study and FP also provided the funding for this work. VTE, WM and DLT are part of the development team of the MMS dataset and helped to integrate it into this work. AP, MOA, DGS, WM, FP, HK, RN, VTE and DLT all helped to discuss and finalise the manuscript.

*Competing interests.*  The authors declare that the research was conducted in the absence of any commercial or financial relationships that could be construed as a potential conflict of interest.

*Acknowledgements.* The work of NG and FP on this study was supported by the German Center for Aviation and Space (DLR) under contract 50 OC 2401. AP was financially supported by the German Center for Aviation and Space (DLR) under contract 50 OC 2201. MOA was supported by UKRI Future Leaders Fellowship MR/X034704/1. This research was supported by the International Space Science Institute (ISSI) in Bern, through ISSI International Team project #546 "Magnetohydrodynamic Surface Waves at Earth's Magnetosphere (and Beyond)". We thank Joe King and Natalia Papitashvili of the National Space Science Data Center (NSSDC) in the NASA/GSFC for the use of the OMNI 2 database. For the purpose of open access, the authors have applied a Creative Commons attribution (CC BY) licence to any Author Accepted Manuscript version arising.

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
