# Peer review of "Investigation of the occurrence of significant deviations in the magnetopause location: Solar wind and foreshock effects"

_EGUsphere, 2024_

## Referee Comment (RC1)

**Manuscript:** Niklas Grimmich et al., egusphere-2024-2956
Investigation of the occurrence of significant deviations in the magnetopause location: Solar wind and foreshock effects.

**General Comments**

The manuscript "Investigation of the occurrence of significant deviations in the magnetopause location: Solar wind and foreshock effects" by Niklas Grimmich et al. contains a huge statistical analysis for the magnetopause location. It combines two empirical models of the magnetopause (Shue et al. (1988), Nguyen et al. (2022)) and data from three different spacecraft missions, CLUSTER, THEMIS and MMS. He tries to study the deviations between the models and the data using different solar wind conditions, different θBn angles and different regions of the magnetosphere. He concludes that foreshock is in some cases responsible for these deviations. From my point of view, this is a work that includes a lot of information and tries to cover all the aspects and possibilities of the deviation issue between the real and the model location of the MP. However, I believe that there are some problems that are not addressed or they are not addressed properly. In this respect, the material presented in this research is valuable and appropriate for publication after a few important points are discussed.

**Scientific Comments/ Major Points**

- You are using Shue et al. (1998) and Nguyen et al. (2022b) models and you explain the reasons behind these choices very well in the discussion part. However, I think it is necessary to mention other models like Lin et al. (2010) and Petrinec and Russel (1996) and the reason why you are **not** using them instead. Aghabozorgi et al. (2023) showed that Lin model is the more accurate for the MP. Also, you are not referring at all to Formisano et al. (1979), who were the first people to examine MP location.
- In Line 25, you refer to "some internal processes". Which processes are those? Can you give some examples to the reader?
- I think that on lines 77-82 that you are referring to the SW plasma categorisation you should put more info (I know you have extensive explanation later), at least for the SW velocity.
- Why are you using above 0.75 probability for the crossings? Why not 0.85 for example? How many crossings did you exclude using this criterion?
- How many crossings did you exclude from your data do to the OMNI criterion you set and why did you choose 8-minute intervals and not 5-minute for example?
- You make some argument between θBn<45 and θBn<60. You have to make clear what you use here (it is visible from the graphs but still). Also, on line 45 you say that errors may not be significant. How can you be certain about this? How did you check it?
- The sketch explains very nicely the deviations but you should mention that Fig. 1 is obviously not on scale.
- Solar Wind separation issues: (1) you do not expect the observations on the boundary to be a problem but this is not right. They can very easily skew your statistics. Maybe you could impose some measure and remove some of them to be sure. (2) why did you use this data classification if you want to make a connection with the foreshock? I believe SW velocity should be enough for classification. The ejecta type is very susceptible to which phase of the ICME you collect the data.
- Mean or median: In your study you decided to use the mean value as an estimator but I believe this makes the study less trustworthy. The median is a way more robust measure, meaning it is not affected by outliers as much as the mean value. The problem is also visible in Table 1, where your interquartile values are huge, for example T(ion) can go negative? This is extremely weird. In line, 173 you have a mistake saying Table 1 shows mean and media values, whereas it shows Median plus/minus the interquartile. Also, I think the "all sources column" make your sample look problematic, where in reality the solar wind measurements are very variable. I am not sure why you included this.

- In Table 2, I think you should consider changing the lines "deviant from…" from absolute numbers to percentages of the original set (equator subsolar, equator flanks etc.).  also, I know you address it later, but if you have only 7 observations this creates a real problem to the results and these results should be explicitly addressed as unreliable. Moreover, a very important question is: when you have many crossings really close to each other of the MP (since MP is dynamic), let's say 6, did you consider all these 6 into your data set or you just chose only one MPC in your data set?
- Figures 2-5: I think that the figures are very small and borderline unreadable. I would prefer 2, 4and 5 to be presented horizontally. In Fig.2 label you write "coloured" for red and blue, but grey is also a colour. Thus, the label needs updating. In Fig. 3 you can write on x-axis the model and on Figs. 4,5 the light purple should become more intense for example magenta or deep purple.
- A large difference between the models is not visible to me. Actually, in many cases the very simple Shue et al. (1998) predicts much better the MPC (equatorial).
- I find the separation dawn/dusk very important due to the MP asymmetry. Could you elaborate a little bit more on why you chose this classification in you text?
- I think there is a mistake in your text (line 264) where you say that Fig.4g, h panels seem to have negative deviations, but I see positive.
- I understand why Figs. 6,7 are important in your analysis but they are so full of information that the reader gets totally lost and they are deemed unreadable. I had to write a separate post-it to understand what I am looking at. I think you should consider some other way to present these results, or restrict some part to the text. The results themselves and the explanation are fine.
- It was my perception in the beginning of the paper that you are going to **compare** the two models, however you just group your results. So, this was not part of the research?
- Additionally, this research is a huge statistical analysis and contains any possibility in terms of position or SW conditions. But I can't find a very strong conclusion. You mention that foreshock **is responsible** for the occurrence of deviant MPCs. I do not exactly agree with this. Foreshock **can be responsible** of course but there are examples of very quiet foreshocks without any transients that are followed by strong MSHs with jets, that can actually travel to the MP and deform it. We can't be sure, or at least we have not proved yet, that foreshock is responsible for what you see. So, I believe you should ease a little bit this conclusion to a strong based argument, for example you can say that the foreshock presence seems to be important but not necessarily crucial and this connection needs further investigation.
- An equivalently strong conclusion is made in lines 401-409. It needs more physical explanation, because I do not see any panel to support your argument.
- In your discussion you do not refer to SRR plasma at all. Could you please add some thoughts?
- Can it be that the models predict the shape of the MP more poorly at the flanks? And if yes, why?

**Technical corrections/ Minor Mistakes**

Abstract
- Lines 2-4: the meaning is clear but the sentence in line 4 is too repetitive.
- Line 3: "in response to" → "with respect to"
- Line 5: … understood, since deviations…
- Line 13: "most frequently" → "more frequently"

Introduction
- Lines 19-21: are exactly the same as lines 1-2. I would suggest that you rephrase one of them.
- Line 31: "undulated", I think you should find a simpler word, "moves" → moving
- Line 31: "due to" (the second) → because of
- Line 40: "due to instabilities" → and instabilities

- Line 44: "another explanation": I can't find which was the first explanation, so maybe you should rephrase. Also, the sentence is too big, maybe you can rephrase like "…from MHD theory. The reduction…"
- Line 45: the subsequent distribution, the magnetosheath, "resulting in" → results in
- Line 52: "a constant motion" → the constant motion
- Line 54: The simplest magnetopause models…, rotational asymmetry does not need article "a"
- Line 56: "…shape as basis, include…" → shape, including
- Line 57: "take into" → taking into
- Line 61: I would rephrase to "These errors, that have the inability to capture the constant motion around the mean location of the magnetopause, could have several causes."
- Line 66: "problems" → complications
- Line 68: "deviation of observed" → deviation of the observed
- Line 69: problem, important mechanisms of the interaction
- Line 71: location, based on
- Line 72-73: wave activity and increased occurrence rate
- Line 74-75: "Thus, one of these overlooked processes could be the modification of the parametres affecting the MP by the formation of…"
- Line 77: Another point of consideration is the…
- Line 84: "too simple" → too simplistic, "the origins of the deviations" → how the deviations originate
- Line 88: and the modelled MP

Datasets and Methods
- Line 91: For our investigation, we…, "collect" → contain
- Line 96: "collected" → include
- Line 98-99: "for details on the coordinate system see" in not necessary, the reference is enough
- Line 101: previous publications
- Line 104: "at a cadence" is not necessary
- Line 106: "crossings is" → crossings are
- Line 151: "might be" → is
- Line 175: "did not" → does not

Results
- Line 190: "to the distributions" is not necessary
- Line 193: "ever" → either
- Line 197-201: I think these lines should be moved after line 183.
- Lines 223-227: Explain the test and these lines seem to be written in a very complex way.
- Line 227: MPCS, $\theta Bn < 45^o$ is a favourable condition.
- Line 229: "a foreshock" → foreshock
- Line 264: "it looked" → looked, "were" → are
- Line 265: "revealed" → reveals

Discussion
- Line 310: Our investigation
- Line 315: "the difference in orbits" → the different orbits
- Line 318: "be even better" → better
- Line 319: previous studies, …
- Line 366: In Figs.
- Line 376: "quasi-radical" → quasi-radial
- Line 399: rather scarce or rather sparse?

Conclusions

- Line 443: "In short" → I don't like this expression, you can use "In summary", "To sum up", "To summarise", three different spacecraft missions, "last two decades" → say the year range
- Line 444: "that occur" is not necessary
- Line 447: "and the" → "when the"
- Line 448: "foreshock is" → you are **not** sure so "foreshock can be"
- Line 450: may lead
- Line 453: in relation with
- Line 457: MPCs may occur

---

## Referee Comment (RC2)

**Referee report on 'Investigation of the occurrence of significant deviations in the magnetopause location: Solar wind and foreshock effects', by Grimmich et al.**

This study investigates possible sources of the scatter (deviations) of observed magnetopause locations about parameterized models of the average magnetopause shape and position. The deviations of the observed magnetopause position from that of the models is investigated as a function of the IMF orientation (i.e., defining the foreshock, leading to convected oscillations), and for different 'types' of solar wind. The cause(s) of deviations of the observed magnetopause location from that expected from statistical models is of interest, and has been examined by several investigators (though not appropriately referenced nor discussed here). Aside from an exploration of different 'types' of solar wind, it is not clear how this investigation improves upon these earlier efforts. There are also many questions about the methodologies employed within this study that need to be answered and comments to be addressed before this study can be considered for publication in Annales Geophysicae.

1)      The schematic of Figure 1 is not even crudely representative of the magnetopause. The standoff position has long been known to be the smallest distance from Earth. Yet, the schematic shows this position to be the furthest distance from Earth and thus, very unrealistic. It would also be helpful to the reader to include in the schematic of Figure 1 the bow shock and angle(s) used.

2)      Lines 146-157: It is confusing to the reader whether $\vartheta_{Bn}$ and $\theta_{Bn}$ are the same or different parameters. Inclusion of the relevant angle(s) in the Figure 1 schematic would be useful. Using the local theta_Bn value at the bow shock model, estimated using the vector normal to the model magnetopause and intersecting the observed magnetopause location, is not a very appropriate method for determining whether or not the spacecraft is downstream of the foreshock (i.e., in the downstream region associated with either the Quasi-parallel or Quasi-perpendicular shock region). When the spacecraft is near the terminator, the 'local' value of theta_Bn can be tens of degrees different from the theta_Bn value near the subsolar shock location; in the vicinity of the foreshock. Especially when the IMF is dominant Bx, or when the IMF Bx component is negligible with respect to the other components, the 'local' theta_Bn can suggest a quasi-parallel region while the magnetopause crossing is actually in a quasi-perpendicular region (or vice versa). Some examples of these regions propagating within the magnetosheath were provided for various IMF orientations by Russell et al., GRL, 663-666, 1983 and also shown in Luhmann et al., JGR, 1711-1715, 1986.

3)      Line 38: Why the adjective 'so-called' foreshock? The existence of the ion and electron foreshock, in multiple planetary systems, has been well-established based on spacecraft observations for decades.

4)      There are several very relevant published studies related to the foreshock and its effects on the magnetopause location and within the magnetosphere that are neglected in the Introduction. Some of these references include:

- As mentioned in point #2, Russell et al., GRL, 663-666, 1983 showed the occurrence rate of Pc 3,4 waves within the inner magnetosphere is much more frequent for small theta_Bn (radial IMF) than for

transverse IMF. Although this study did not explicitly examine magnetopause deviations, it was postulated that magnetospheric ULF wave activity is associated with Kelvin-Helmholtz waves along the magnetopause as a consequence of convected foreshock activity.

• Luhmann et al., JGR, 1711-1715, 1986 examined transverse and compressional wave activity within the magnetosheath as a function of the IMF configuration and local time. The result of this study also implied that compressional and transverse oscillations originating upstream convect through the magnetosheath and affect the magnetopause location.

• Song et al., GRL, 744-747, 1988 described the magnetopause oscillation amplitude as a function of IMF configuration (their Table 1) and distance downtail (solar zenith angle).

• Russell et al., GRL, 1439-1441, 1997 showed a significant statistical dawn/dusk difference in observed multiple magnetopause boundary crossings (per pass) and average oscillation amplitude, attributed to convected foreshock effects. Differences as a function of IMF clock angle were also noted. Petrinec et al., JGR, 2022, doi:10.1029/2021JA029669 also observed very similar multiple magnetopause crossing statistics, consistent with convected oscillations from the foreshock region.

5)      Lines 66-69: How can a *quantitative* assessment of the percentage of cases of significant magnetopause location deviations be attributed to foreshock effects, when there are multiple other parameters that are known to affect the average location; but are not accounted for in the models? In addition to those listed in the manuscript, some examples of neglected parameters include:

• The Region 1 current strength (Sibeck et al., JGR, 5489, 1991), also expressed through the ring current effect (Dst*) (e.g., Hayosh et al., Adv. Space Res., 2417-2422, 2005; Machkova et al., JGR, 905-914, 2019). This can affect the average magnetopause location by a few tenths of an RE.

• Earth's lowest-order magnetic moment is actually best described by an offset dipole (Laundal and Richmond). At the distance of the magnetopause, only this moment survives (higher order moments decrease much more rapidly with increasing distance from Earth). The offset is ~577 km, which translates into calculable variation of the magnetopause subsolar distance of up to ~±0.1 RE (depending on season (dipole tilt) and time of day of the crossings). This was shown in the empirical study of Machkova et al., JGR, 905-914, 2019.

• In addition, the average models do not capture the time history of the solar wind. The history can greatly affect the magnetopause location due to ongoing processes such as erosion due to reconnection.

• It is commended that Kelvin Helmholtz waves were mentioned; even if just briefly. It would be more helpful if there were a quantitative assessment of the contribution of KH to 'deviant' magnetopause crossings within the four dayside magnetopause regions (even if the instability is not fully developed (e.g., Hasegawa et al., JGR, 2003, doi:10.1029/2002JA009667; Henry et al., JGR, 11888-11900, 2017; Radhakrishnan et al., JGR, 2024, doi:10.1029/2024JA032869)), or at least an estimated assessment of the relative contribution of KH to that of convected foreshock oscillations in relation to 'deviant' magnetopause crossing locations.

6)      Figs., 2,4,5 caption: The captions mention a 'reported 1 RE uncertainty'. Where has this number been reported, and why is it constant for all four magnetopause regions? It's typically understood that the

model uncertainty increases further away from the standoff point. While Shue et al. 1997 report a single value of 1.24 RE standard deviation between model and observations (their Fig.15), it's shown in their figure that there is increased scatter for larger magnetopause distances (typically flanks) when compared to smaller distances (typically standoff region). Shue et al. 1998 reported a standard deviation of 1.23 RE. The uncertainties of the individual fit coefficients {$a_n$} should provide a more appropriate estimate of the magnetopause uncertainty in each of the four magnetopause regions. It may be that an uncertainty value of ~1 RE is reasonable for the dayside magnetopause; but it needs to be justified with a specific reference and/or an explicit calculation.

7)      Lines 98-99: Although the GSE coordinate system is described in Laundal and Richmond, SSR, 2017, there is no description of aberration. Please describe whether the aberration as used in this study is a fixed angle applied to all observed crossings, or uses the actual measured solar wind speed for each magnetopause crossing, or uses the full solar wind velocity (all components) in the calculation of the aberration angles.

8)      Line 134: It is stated that the Nguyen et al. 2022 model (N22b) is an extension of the SH98 model. Although the basic zenith angle functional form is the same, this model is quite different. The IMF Bz dependence of the N22b magnetopause standoff distance is very different from Shue et al. 1998. This N22b dependence on IMF Bz does not match what has been observed and described over decades of empirical magnetopause studies (including those by one of this manuscript's co-authors). Specifically, the erosion of the dayside magnetosphere (as documented by $\Delta r_{0mp}$) for a given value of southward IMF Bz has long been known to be much greater than the expansion of the dayside magnetosphere for an equivalent value (but opposite sign) of northward IMF Bz.

9)      Lines 185-189: The treatment of orbital bias in the statistical analysis is curious. It appears that the authors are trying to weight the sampling of magnetopause regions so that rarely sampled regions have equal representation (coverage) with those regions that are more often sampled. If this is the case, then this is a different type of orbital bias than is normally of concern. Especially for studies of the average magnetopause location (and deviations from the average shape), the orbital bias of concern is primarily due to spacecraft apogees which are lower than the average boundary location; so the spacecraft can only sample the magnetopause during the innermost transient excursions, or for intervals of high solar wind pressure. For example, the THEMIS A,D,E missions only have apogees of ~13.2-13.7 RE, while the nominal magnetopause location near the terminator is ~14.5 RE. Similarly, the MMS spacecraft during the prime mission had an apogee of 12 RE; and so could only rarely and briefly sample the magnetopause a few hours away from local noon. Because of this small MMS apogee during the prime mission, those magnetopause crossings shouldn't be used in regions where they cannot adequately sample at least the average boundary location, for determination of the general magnetopause shape. It is very important to also address this orbital bias, and how it affects the statistical results of this study.

10)     Lines 194-196, Lines 343-344: The significant skewing of the distributions relative to the N22b model suggests that either the functional form used for their model is not appropriate (cf., point #8), and/or the data set used to fit their model is afflicted by the orbital bias (i.e., limited spacecraft apogees) that is described in point #9. This should be addressed.

11)      Fig.3: Are these histograms normalized such that each histogram distribution at theta_Bn = 45 degrees is set to '1'?

12)      Figs.4,5, and lines 258-259: Are the magnetopause models including the aberration of the solar wind? If not, that would explain the differences between dawn and dusk between the observed and model magnetopause locations (orbital bias of limited apogee (e.g., THEMIS A,D,E) may also contribute to the observed dawn/dusk differences). The models and observations should be consistent with one another (i.e., in the same coordinate system). Please provide additional description.

13)      There are several sentences in Sections 3 and 4 which are too convoluted and ambiguous for the reader to understand. This (along with several other issues described in this report) strongly suggests that the co-authors have not read this manuscript. A few examples include Lines 264-269; Lines 332-334; Lines 371-372.

Some additional minor comments:

- Lines 70-72: Re-arrange and separate this into two sentences.

"For example, high solar wind speeds appear to lead to an anti-Earthward expansion and outward displacement of the MP from the predicted model location, which is based on the assumption that the higher dynamic pressure in these cases compresses the MP (Grimmich et al., 2023a, 2024b)."

->

For example, there is an assumption that higher dynamic pressure compresses the MP. However, high solar wind speeds appear to lead to an anti-Earthward expansion and outward displacement of the MP from the predicted model location (Grimmich et al., 2023a, 2024b).

- Line 72: 'foreshock, is' -> 'foreshock is'
- Table 1, 1st column: kms^-1 -> km s^-1
- Table 1 footnote: 'to the the four paramters' -> 'to the four parameters'
- Line 181: 'a classification results from' -> 'a classification from'
- Table 2 caption, line 2: 'in the' -> 'into the'
- Line 194: ever -> either
- Line 197: 'In a next step,' -> 'For the next step,'
- Line 210: indention -> indentation
- Line 217: cause -> course
- Fig.4 caption, 2nd to last line: 'a associated' -> 'an associated'
- Fig.4 caption, last line: line -> lines
- Line 276: 'different magnetospheric regions' -> 'four different magnetopause regions'
- Line 283: 'with EJC plasma' -> 'with only EJC plasma'
- Line 315: 'in the table 2 which' -> 'in table 2, which'
- Line 322: cavities -> deficiencies

- Line 323: 'was include here.' -> 'was included here.'
- Line 325: 'cusp indents' -> 'cusp indentations'
- Line 338: 'point out the similar errors of empirical modelling' -> 'point out similar uncertainties inherent in the empirical modelling'
- Line 376: quasi-radical -> quasi-radial
- Line 377: 'the foreshock develops directly in front of the bow shock nose,'. This is not strictly true. The point of 'attachment' of the foreshock to the bow shock surface is not necessarily the nose of the bow shock; but wherever the IMF is tangent to the bow shock surface. Please reword.
- Line 391: 'from a combined datasets' -> 'from combined datasets'
- Lines 408-409: 'the EJC plasma composition': What does this mean? Is this the general characteristics of Table 1? Or does this refer to a higher percentage of heavy ions (e.g., alphas)? Or does this refer to a composition that includes a higher than normal 'hot' plasma content? Please be specific.
- Line 420: indention -> indentation
- Line 429: errors -> uncertainties
- Lines 433-434: '(SMILE) mission … will again be a near-Earth, polar-orbiting satellite,'. This isn't right. The SMILE mission is rather high inclination (70° or 98°); but not a polar mission (inclination of 90°). It's also not in a near-Earth orbit: It's to go into a highly-elliptical orbit with perigee of ~1.8 RE (5000 km altitude), and apogee of ~20 RE.
- Line 453: 'relation the foreshock' -> 'relation to the foreshock'
- Lines 453-454: As described earlier, the Song et al. 1988 study examined the amplitude of MP motion, and the Russell et al. 1997 study already examined this amplitude in relation to the foreshock (IMF orientation and local time). How would this future study be different?
- Line 468: 'study FP' -> 'study and FP'

---

## Author Comment (AC1)

[revised manuscript text omitted]

---

## Author Response (AR1)

**Author Response to Reviewer comments**

First of all, we would like to thank both reviewers for their time and effort in reviewing our paper and for their good and helpful comments. These have helped us to improve our paper significantly. In the following we response to all the comments raised by the reviewers.

This is a compilation of all *the comments from the reviewers* and our responses to them, as we have posted them on the public discussion page. We have not listed here all the technical/language corrections suggested by both reviewers, but have incorporated them all into our updated manuscript. Nonetheless, the minor comments made by the referees, which go beyond simple language changes, are appropriately addressed below.

**Scientific Comments/ Major Points of Referee #1:**

*You are using Shue et al. (1998) and Nguyen et al. (2022b) models and you explain the reasons behind these choices very well in the discussion part. However, I think it is necessary to mention other models like Lin et al. (2010) and Petrinec and Russel (1996) and the reason why you are not using them instead. Aghabozorgi et al. (2023) showed that Lin model is the more accurate for the MP. Also, you are not referring at all to Formisano et al. (1979), who were the first people to examine MP location.*

> We thank the reviewer for their suggestions and have included the mentioned references to other models in the introduction of our update manuscript. Additionally, we have added some discussion of why we did not use the Petrinec and Russell model (because its initial indentation is designed to fit the magnetotail MP and not the dayside) or the Lin et al. model (because the cusp indentation are more accurately depicted in the N22b model).

*In Line 25, you refer to "some internal processes". Which processes are those? Can you give some examples to the reader?*

> We have added references to papers by Sibeck et al. (1991) and Machková et al. (2019) in our introduction, which discuss the influences of the ring and Birkland currents on MP location, to give some examples of the internal processes.

*I think that on lines 77-82 that you are referring to the SW plasma categorisation you should put more info (I know you have extensive explanation later), at least for the SW velocity.*

> We have followed the reviewer's suggestion and included information on the solar wind velocities of the different plasma types in our introduction. Our main explanation of the four types can still be found in the later parts of our manuscript.

*Why are you using above 0.75 probability for the crossings? Why not 0.85 for example? How many crossings did you exclude using this criterion?*

> We excluded 25% of the data in the THEMIS dataset and 35% of the data in the Cluster dataset. Our threshold was set in the study by Grimmich et al. (2023) and was more or less

arbitrarily chosen. However, it seems to be a good threshold to minimise the inclusion of false identifications in the dataset, while not scraping too many of the identifications.

*How many crossings did you exclude from your data do to the OMNI criterion you set and why did you choose 8-minute intervals and not 5-minute for example?*

The OMNI criteria excluded about 28% of the original data. The 8-minute interval for the OMNI data was already used before in the studies by Grimmich et al.. Especially in Grimmich et al. (2024) it was discused why this interval was choosen: *"Various solar-wind parameters from the OMNI dataset are taken as the mean values in an 8 min interval preceding the crossing if up to 75 % of the data points are available in that interval. The length of the interval chosen takes into account the time delay from the bow shock to the MP and terminator. To estimate this time delay, we assumed typical distances between the bow shock and the MP of 3 to 4 RE, a typical distance from the MP to the terminator of 10 RE, and typical flow velocities in the subsolar (flank) magnetosheath of 100 km s−1 (300 km/s). Using these numbers to calculate how long it would take a plasma element to travel from the BS to the MP and then to the terminator gives the 8 min we considered to be the time delay."* In our update text, we refer to this publication and the discussion in it.

*You make some argument between $\theta Bn<45$ and $\theta Bn<60$. You have to make clear what you use here (it is visible from the graphs but still). Also, on line 45 you say that errors may not be significant. How can you be certain about this? How did you check it?*

Due to the second referee's comment, we have included another method to determine the bow shock configuration for each crossing. By comparing the two methods we got a agreement of 80%. We now use both methods for the analysis, which minimises the errors in the angle calculation and should make our results slightly more reliable than before. In addition, we make it clear in our text which threshold we use to identify the foreshock.

*The sketch explains very nicely the deviations but you should mention that Fig. 1 is obviously not on scale.*

We agree with the reviewer and have added an explanation of the scale to the caption of Fig. 1. Overall, we have also updated the figure to show the bow shock model and the relevant angles used in our study.

*Solar Wind separation issues: (1) you do not expect the observations on the boundary to be a problem but this is not right. They can very easily skew your statistics. Maybe you could impose some measure and remove some of them to be sure. (2) why did you use this data classification if you want to make a connection with the foreshock? I believe SW velocity should be enough for classification. The ejecta type is very susceptible to which phase of the ICME you collect the data.*

On point (1): As suggested by the reviewer, we have introduced a method to remove edge cases that might skew our statistics. The original classification compares 3 different values computed from the solar wind parameters, whether they are above or below 1. For each class, we have applied a second threshold close to the one that needs to be statistically

established for a more reliable classification result. These new thresholds have been chosen
by looking at the classification of our 24 years of OMNI data in such a way that about 5% of
each class is marked as an ambiguous edge case. We have included an explanation of this
in our update manuscript at the end of section 2.

One point (2): The purpose of our study was not only to make a connection to the
foreshock, but also to see if not only single parameters (as we have done it in previous
studies), but parameter combinations of the solar wind are often associated with deviant
MPC. The classification scheme is an appropriate tool for this purpose and we have
therefore included it in this study.

*Mean or median: In your study you decided to use the mean value as an estimator but I believe
this makes the study less trustworthy. The median is a way more robust measure, meaning it is
not affected by outliers as much as the mean value. The problem is also visible in Table 1, where
your interquartile values are huge, for example T(ion) can go negative? This is extremely weird. In
line, 173 you have a mistake saying Table 1 shows mean and media values, whereas it shows
Median plus/minus the interquartile. Also, I think the "all sources column" make your sample
look problematic, where in reality the solar wind measurements are very variable. I am not sure
why you included this.*

To give our study more confidence, we have now redone our calculation using the median
of the 8 min OMNI intervals as an estimator for the solar wind conditions applied to the
MP.

We also agree with the reviewer that some of the values in our table 1 are strange.
While IQR (the difference between 25th and 75th percentiles) was used, it was incorrectly
expressed as a plus/minus around the median. This does not take into account potential
skewness in the distribution, leading to the confusion. The median absolute deviation
measures the symmetric deviation from the median of the distribution, which is more
appropriate to express as a plus/minus measure of spread. We now use these median
absolute deviations as a measure of the spread of the parameter distributions, which
shows much more reasonable ranges. We have updated our manuscript accordingly.

In addition, we do not think that we should exclude the "all sources" column, as it serves
as an average against which we can compare the parameter settings of the different types.

If this comment was also about our use of the mean in Figure 2-5, we would like to clarify
that these mean values come from the Gaussian fits we have applied and are therefore
identical to a median value. We are aware that the fitted value is not identical to the true
mean or median of the data collected. However, using the fitted values for all distributions
gives us a more reliable value to compare.

*In Table 2, I think you should consider changing the lines "deviant from…" from absolute
numbers to percentages of the original set (equator subsolar, equator flanks etc.). also, I know you
address it later, but if you have only 7 observations this creates a real problem to the results and
these results should be explicitly addressed as unreliable. Moreover, a very important question is:*

125 *when you have many crossings really close to each other of the MP (since MP is dynamic), let's say*
*6, did you consider all these 6 into your data set or you just chose only one MPC in your data set?*
        We thank the reviewer for their suggestions regarding Table 2, which we have incorporated
        into our manuscript. As part of these changes, we have also changed the heading of the
        table and added a footnote to indicate the unreliable subsets. Regarding the question of
130     multiple crossings, we consider all crossings in the dataset, i.e. we have not filtered out
        multiple crossings that are really close to each other. We have added this information to
        our results section.

*Figures 2-5: I think that the figures are very small and borderline unreadable. I would prefer 2,*
135 *4and 5 to be presented horizontally. In Fig.2 label you write "coloured" for red and blue, but grey*
*is also a colour. Thus, the label needs updating. In Fig. 3 you can write on x-axis the model and on*
*Figs. 4,5 the light purple should become more intense for example magenta or deep purple.*
        Following the reviewer's suggestion, we have changed the orientation of Figures 2, 4 and 5,
        updated the caption of Figure 2 to make it clear which coloured region we are referring to,
140     and added the information about the models to the x-axis of Figure 3. We have also tried,
        where possible, to match the text size of the figures for better readability. Our updated
        figures are attached to these responses.

*A large difference between the models is not visible to me. Actually, in many cases the very simple*
145 *Shue et al. (1998) predicts much better the MPC (equatorial).*
        We are in agreement with the reviewer on this point. The Shue et al. model on average
        performs very well in the equatorial regions. Overall, the Nguyen et al. model seems to
        have some problems with underestimating the MP location, which we have further
        addressed in our update manuscript. In general, the fact that there were no major
150     differences between the models leads us to believe that our results are more reliable and
        that there are indeed some effects that could/should be included in future models.

*I find the separation dawn/dusk very important due to the MP asymmetry. Could you elaborate a*
*little bit more on why you chose this classification in you text?*
155     We agree with the reviewer that this separation is important. Our reasoning for this
        separation was that, in the Parker spiral IMF, the foreshock is often at dawn and we
        expected to see some asymmetry that we might be able to discuss further with our results.
        This separation also allowed us to divide our data into even more subsets, so that we could
        see more clearly in which regions the models were predicting reasonably well and in which
160     regions, for example, shock effects were taking hold. This has been incorporated into our
        update manuscript with regard to the discussion of Figure 5.

*I think there is a mistake in your text (line 264) where you say that Fig.4g, h panels seem to have*
*negative deviations, but I see positive.*
165     We disagree, panels g and h of Figure 4 show lower mean values for the distribution
        associated with the foreshock, and therefore a shift towards negative deviations. However,

we are rewriting these sections based on other comments from reviewers, so that this may be less confusing in our updated manuscript version.

170 *I understand why Figs. 6,7 are important in your analysis but they are so full of information that the reader gets totally lost and they are deemed unreadable. I had to write a separate post-it to understand what I am looking at. I think you should consider some other way to present these results, or restrict some part to the text. The results themselves and the explanation are fine.*

We agree with the reviewer that the figures referred to are not easy to read. We have
175 simplified Figures 6 and 7 into a new Figure 6 and moved the information on the IMF orientation to a separate tables in our updated manuscript. The new figure and the table from our manuscript are attached to this response.

*It was my perception in the beginning of the paper that you are going to compare the two
180 models, however you just group your results. So, this was not part of the research?*

We have compared the models, you can see this in our results section where we have always shown and highlighted the values from both models. However, it is true that we did not really discuss the differences and similarities between the models. We have added a paragraph to our discussion section to discuss the models a little further. We also
185 concluded that the SH98 model was generally better at predicting MP location than the N22b model, although bias may still be obscuring our results.

*Additionally, this research is a huge statistical analysis and contains any possibility in terms of position or SW conditions. But I can't find a very strong conclusion. You mention that foreshock
190 is responsible for the occurrence of deviant MPCs. I do not exactly agree with this. Foreshock can be responsible of course but there are examples of very quiet foreshocks without any transients that are followed by strong MSHs with jets, that can actually travel to the MP and deform it. We can't be sure, or at least we have not proved yet, that foreshock is responsible for what you see. So, I believe you should ease a little bit this conclusion to a strong based argument, for example you
195 can say that the foreshock presence seems to be important but not necessarily crucial and this connection needs further investigation.*

We agree with the reviewer that our results do not provide a definitive answer and that we may need to temper our conclusions. We have expanded our discussion section quite a bit to point out some of the problems with our results that were commented on by the second
200 reviewer. For example, we now discuss that some of the anomalous MPCs thought to be caused by the foreshock may actually be caused by KHI, and also that bias may still be affecting our statistics. However, it is well established that foreshock and/or quasi-parallel shock processes are responsible for the MSH jets. Thus, in general, our statistics show that foreshock/quasi-parallel shock conditions are conducive to anomalous MPCs, even if they
205 are not directly caused by the foreshock itself. By adding this consideration to our manuscript, we have also updated our conclusion section to incorporate the new discussions.

*An equivalently strong conclusion is made in lines 401-409. It needs more physical explanation,*
*because I do not see any panel to support your argument.*

> Due to the revision of the figures and some changes to the results, we have rephrased parts of the Results and Discussion section. In particular, this part has been expanded and rephrased in the updated manuscript to soften the conclusion we draw.

*In your discussion you do not refer to SRR plasma at all. Could you please add some thoughts?*

> We disagree with the reviewer on this point. We did mention SRR plasma in our manuscript, but only briefly: "*Contrary to the finding of Grimmich et al. (2023), which suggests that high Alfvén Mach numbers and solar wind plasma ß are also important for the occurrence of extended MPCs, we find that the SRR plasma described by these conditions is less important and actually only relevant together with the radial IMF, which is likely to be the dominant effect for the occurrence of deviant MPCs.*"

*Can it be that the models predict the shape of the MP more poorly at the flanks? And if yes, why?*

> Yes, it is possible that the MP shape is worse predicted at the flanks. One effect could be an orbital bias due to spacecraft apogee, which is less than the average MP location on the flanks. As a result, the models may have underpredicted the actual position of the MPs on the flanks. Another point could be that due to KHI the flank MP is very often in motion and therefore there will be a lot of scatter around the average MP shape, which could lead to ambiguous fit results from models. We have included these points in the discussion of our results.

**Additional Minor Points of Referee #1:**

*Lines 2-4: the meaning is clear but the sentence in line 4 is too repetitive.*
*Lines 19-21: are exactly the same as lines 1-2. I would suggest that you rephrase one of them.*

> We have addressed these two comments in our updated manuscript by removing the first line of the abstract. Thus, repetition should no longer be an issue.

*Line 197-201: I think these lines should be moved after line 183,*

> We agree with this suggestion and have moved the paragraph after the line mentioned. As a result of this move, we have also rephrased some parts of the sentence.

*Lines 223-227: Explain the test and these lines seem to be written in a very complex way.*

> We have expanded and rephrased the paragraph in our update manuscript to further explain the statistical test we used. In full, our new paragraph now reads: "*..., we performed a Mann-Whitney U test. This test is a generalisation of Student's t-test for non-normal distributions like ours and is used to analyse the differences between two independent samples with similar shapes. The null hypothesis of the test is always that there are no significant differences between the samples, which in our case would mean that the distribution of deviant events is the same as the reference distribution. To reject or accept the null hypothesis, a rank is assigned to each observation. The ranks are then summed for each group. The test statistic U-value is calculated from these rank sums for each distribution and the smaller of the U-values is used for hypothesis testing. If the*

*calculated U-value is less than the critical value from the Mann-Whitney distribution (based on sample sizes), the null hypothesis is rejected, indicating a significant difference between the two groups. In our case, the probability value from the test statistic must be smaller than 0.05 (see Mann and Whitney, 1947 for more details)."*

255

**Scientific Comments/ Major Points of Referee #2:**

*The schematic of Figure 1 is not even crudely representative of the magnetopause. The standoff position has long been known to be the smallest distance from Earth. Yet, the schematic shows this position to be the furthest distance from Earth and thus, very unrealistic. It would also be*

260 *helpful to the reader to include in the schematic of Figure 1 the bow shock and angle(s) used.*

We agree with the reviewer that it might be helpful to include the bow shock in our sketch, so we have included it and the relevant parameters around it in Fig. 1 (see attached Fig). In modifying the sketch, we have also tried to make the MP more realistic and shortened the stand-off distance. The modification of the sketch has led to some rewriting in the caption.

265

*Lines 146-157: It is confusing to the reader whether $\vartheta Bn$ and $\theta Bn$ are the same or different parameters. Inclusion of the relevant angle(s) in the Figure 1 schematic would be useful. Using the local theta_Bn value at the bow shock model, estimated using the vector normal to the model magnetopause and intersecting the observed magnetopause location, is not a very appropriate*

270 *method for determining whether or not the spacecraft is downstream of the foreshock (i.e., in the downstream region associated with either the Quasi-parallel or Quasi-perpendicular shock region). When the spacecraft is near the terminator, the 'local' value of theta_Bn can be tens of degrees different from the theta_Bn value near the subsolar shock location; in the vicinity of the foreshock. Especially when the IMF is dominant Bx, or when the IMF Bx component is negligible*

275 *with respect to the other components, the 'local' theta_Bn can suggest a quasi-parallel region while the magnetopause crossing is actually in a quasi-perpendicular region (or vice versa). Some examples of these regions propagating within the magnetosheath were provided for various IMF orientations by Russell et al., GRL, 663-666, 1983 and also shown in Luhmann et al., JGR, 1711-1715, 1986.*

280 The angle expressions should be the same and we correct our typo. We have also added the angle geometry to our Figure 1 to avoid further confusion. On the point of estimating the shock range: We agree with the reviewer that our method may be flawed and may not always determine the correct configuration for an observed MPC. To validate our calculations, and also to minimise the risk of false classifications, we used the method

285 found in the paper by Petrinec et al. (2022) suggested by the reviewer, which uses the spacecraft position and IMF vectors to determine which shock configuration the spacecraft is behind from a parameter between -1 and 1. This parameter takes advantage of the fact that the IMF cone angle is identical to $\vartheta_{Bn}$ in the sub-solar magnetosphere, and maps this cone angle to other regions with the difference between the spacecraft clock angle and the

290 IMF clock angle to the corresponding position of the observation. Explicitly

$$q = \cos(cone\_angle)*\cos(\phi_{yz\text{-}IMF} - \phi_{yz}) = (B_x/B_T)*\cos(\tan^{-1}(B_y/B_z) - \tan^{-1}(y/z)_{SC}) \text{ is used.}$$

Both methods agree in their predictions up to 80% of the time, and we used only these overlapping predictions for our statistics in the update manuscript. This change has resulted in an additional paragraph in our Methods section and some rewording in the Results and Discussion sections, where we have also incorporated the suggested references.

*Line 38: Why the adjective 'so-called' foreshock? The existence of the ion and electron foreshock, in multiple planetary systems, has been well-established based on spacecraft observations for decades.*

We included the adjective in the first place for readers who may not be very familiar with the region. However, as the reviewer rightly points out, it may not be necessary, so we have removed it from the sentence.

*There are several very relevant published studies related to the foreshock and its effects on the magnetopause location and within the magnetosphere that are neglected in the Introduction. Some of these references include:*

*As mentioned in point #2, Russell et al., GRL, 663-666, 1983 showed the occurrence rate of Pc 3,4 waves within the inner magnetosphere is much more frequent for small theta_Bn (radial IMF) than for transverse IMF. Although this study did not explicitly examine magnetopause deviations, it was postulated that magnetospheric ULF wave activity is associated with Kelvin-Helmholtz waves along the magnetopause as a consequence of convected foreshock activity.*

*Luhmann et al., JGR, 1711-1715, 1986 examined transverse and compressional wave activity within the magnetosheath as a function of the IMF configuration and local time. The result of this study also implied that compressional and transverse oscillations originating upstream convect through the magnetosheath and affect the magnetopause location.*

*Song et al., GRL, 744-747, 1988 described the magnetopause oscillation amplitude as a function of IMF configuration (their Table 1) and distance downtail (solar zenith angle).*

*Russell et al., GRL, 1439-1441, 1997 showed a significant statistical dawn/dusk difference in observed multiple magnetopause boundary crossings (per pass) and average oscillation amplitude, attributed to convected foreshock effects. Differences as a function of IMF clock angle were also noted.*

*Petrinec et al., JGR, 2022, doi:10.1029/2021JA029669 also observed very similar multiple magnetopause crossing statistics, consistent with convected oscillations from the foreshock region.*

We thank the reviewer for bringing these studies to our attention. We have included the suggested reference in our manuscript. In particular, in the introduction we have devoted a paragraph to the oscillation from the foreshock and its effect on the magnetopause. We have also referred to this publication in our discussions and conclusions, which expand on both sections.

*Lines 66-69: How can a quantitative assessment of the percentage of cases of significant magnetopause location deviations be attributed to foreshock effects, when there are multiple*

*other parameters that are known to affect the average location; but are not accounted for in the*
335 *models? In addition to those listed in the manuscript, some examples of neglected parameters include:*

*The Region 1 current strength (Sibeck et al., JGR, 5489, 1991), also expressed through the ring current effect (Dst\*) (e.g., Hayosh et al., Adv. Space Res., 2417-2422, 2005; Machkova et al., JGR, 905-914, 2019). This can affect the average magnetopause location by a few tenths of an RE.*

340 *Earth's lowest-order magnetic moment is actually best described by an offset dipole (Laundal and Richmond). At the distance of the magnetopause, only this moment survives (higher order moments decrease much more rapidly with increasing distance from Earth). The offset is ~577 km, which translates into calculable variation of the magnetopause subsolar distance of up to ~±0.1 RE (depending on season (dipole tilt) and time of day of the crossings). This was shown in the*
345 *empirical study of Machkova et al., JGR, 905-914, 2019.*

*In addition, the average models do not capture the time history of the solar wind. The history can greatly affect the magnetopause location due to ongoing processes such as erosion due to reconnection.*

*It is commended that Kelvin Helmholtz waves were mentioned; even if just briefly. It would be*
350 *more helpful if there were a quantitative assessment of the contribution of KH to 'deviant' magnetopause crossings within the four dayside magnetopause regions (even if the instability is not fully developed (e.g., Hasegawa et al., JGR, 2003, doi:10.1029/2002JA009667; Henry et al., JGR, 11888-11900, 2017; Radhakrishnan et al., JGR, 2024, doi:10.1029/2024JA032869)), or at least an estimated assessment of the relative contribution of KH to that of convected foreshock oscillations*
355 *in relation to 'deviant' magnetopause crossing locations.*

We would like to thank the reviewer for pointing out other interesting publications that we were not aware of. Some of the additional references listed here have been incorporated into the introduction and discussion of our study. We know that we neglected effects in our study which might also contribute to the deviations. To account for some of the above
360 effects, we implemented the model correction for the ring current effect from Machková et al. (2019) in our study.

The other points raised by the reviewer, such as time history or KH, are now addressed in more detail in our updated discussion section. On the topic of KH assessment, we feel that this would add a lot to this already extensive publication and have therefore not included it
365 directly in our manuscript. However, we are planning some follow-up studies that will address this issue, and we mention this in our conclusion.

*Figs., 2,4,5 caption: The captions mention a 'reported 1 RE uncertainty'. Where has this number been reported, and why is it constant for all four magnetopause regions? It's typically understood*
370 *that the model uncertainty increases further away from the standoff point. While Shue et al. 1997 report a single value of 1.24 RE standard deviation between model and observations (their Fig.15), it's shown in their figure that there is increased scatter for larger magnetopause distances (typically flanks) when compared to smaller distances (typically standoff region). Shue et al. 1998 reported a standard deviation of 1.23 RE. The uncertainties of the individual fit coefficients {an}*
375 *should provide a more appropriate estimate of the magnetopause uncertainty in each of the four*

*magnetopause regions. It may be that an uncertainty value of ~1 RE is reasonable for the dayside magnetopause; but it needs to be justified with a specific reference and/or an explicit calculation.*

This 1 RE uncertainty is mainly based on the results of Staples et al. (2020). This paper examines the agreement between the SH98 model and spacecraft observations on the equatorial dayside and finds that "*...for the majority of events (74%), the magnetopause model can be used to estimate magnetopause location to within ±1 RE.*" Another paper claiming that the SH98 model has an average uncertainty of 1 Re is the work of Case and Wild (2013). We have included references to this paper in the caption of the first figure mentioning this reported error to justify our statement.

*Lines 98-99: Although the GSE coordinate system is described in Laundal and Richmond, SSR, 2017, there is no description of aberration. Please describe whether the aberration as used in this study is a fixed angle applied to all observed crossings, or uses the actual measured solar wind speed for each magnetopause crossing, or uses the full solar wind velocity (all components) in the calculation of the aberration angles.*

To calculate the aberration, we use the measured solar wind speed (the magnitude) from the OMNI dataset for each crossing. If no data are available, we use an average velocity of 400 km/s for the calculation. This together with a fixed value of 30 km/s for the orbital velocity of Earth results in mostly individual aberration angles for each crossing. We added this information to the updated manuscript.

*Line 134: It is stated that the Nguyen et al. 2022 model (N22b) is an extension of the SH98 model. Although the basic zenith angle functional form is the same, this model is quite different. The IMF Bz dependence of the N22b magnetopause standoff distance is very different from Shue et al. 1998. This N22b dependence on IMF Bz does not match what has been observed and described over decades of empirical magnetopause studies (including those by one of this manuscript's co-authors). Specifically, the erosion of the dayside magnetosphere (as documented by ✏️romp) for a given value of southward IMF Bz has long been known to be much greater than the expansion of the dayside magnetosphere for an equivalent value (but opposite sign) of northward IMF Bz.*

We have incorporated this criticism from the reviewer into our manuscript. In our introduction to the functional form of the models in the Methods section, we have rephrased some sentences so that they now read as follows "*This [N22b] model incorporates the above features, and while similar to the SH98 model in its basic zenith angle function, the IMF dependence in the MP stand-off distance is different, showing a weaker effect on MP motion under changing IMF. The functional form of the N22b model is described by ...*"

*Lines 185-189: The treatment of orbital bias in the statistical analysis is curious. It appears that the authors are trying to weight the sampling of magnetopause regions so that rarely sampled regions have equal representation (coverage) with those regions that are more often sampled. If this is the case, then this is a different type of orbital bias than is normally of concern. Especially for studies of the average magnetopause location (and deviations from the average shape), the*

*orbital bias of concern is primarily due to spacecraft apogees which are lower than the average boundary location; so the spacecraft can only sample the magnetopause during the innermost transient excursions, or for intervals of high solar wind pressure. For example, the THEMIS A,D,E missions only have apogees of ~13.2-13.7 RE, while the nominal magnetopause location near the terminator is ~14.5 RE. Similarly, the MMS spacecraft during the prime mission had an apogee of 12 RE; and so could only rarely and briefly sample the magnetopause a few hours away from local noon. Because of this small MMS apogee during the prime mission, those magnetopause crossings shouldn't be used in regions where they cannot adequately sample at least the average boundary location, for determination of the general magnetopause shape. It is very important to also address this orbital bias, and how it affects the statistical results of this study.*

> We would like to thank the reviewer for bringing this to our attention and we believe that this bias may indeed still be present in our data. This may explain why we observe drastically more compressed MPCs than expanded MPCs on the flanks. We mention and discuss this issue in a newly added paragraph at the beginning of our Discussion section.

*Lines 194-196, Lines 343-344: The significant skewing of the distributions relative to the N22b model suggests that either the functional form used for their model is not appropriate (cf., point #8), and/or the data set used to fit their model is afflicted by the orbital bias (i.e., limited spacecraft apogees) that is described in point #9. This should be addressed.*

> We agree with the reviewer and thank them for their insight into what could be causing our distributions to be skewed. We have updated our statement on the origin of the N22b distribution skew and addressed the points raised in the revised manuscript.

*Fig.3: Are these histograms normalized such that each histogram distribution at theta_Bn = 45 degrees is set to '1'?*

> No, as discribed in the text: "*The distributions of $\vartheta_{B,n}$ associated with the outlying MPCs have been normalised by dividing these distributions by the reference distribution, which includes all times when a given $\vartheta_{B,n}$ value is observed and is not restricted to times when MPCs are observed.*" To clarify, the reference distribution is the distribution of each occurrence of a given value at a random location over the time ranges studied, and the second is the distribution of the angles associated with the MPC observations. Rather than directly comparing these two distributions in a graph, we divided the MPC-associated distribution by the reference to see in which parameter ranges the distribution deviates from each other, which would be indicated by values above one in our Fig. 3.

*Figs.4,5, and lines 258-259: Are the magnetopause models including the aberration of the solar wind? If not, that would explain the differences between dawn and dusk between the observed and model magnetopause locations (orbital bias of limited apogee (e.g., THEMIS A,D,E) may also contribute to the observed dawn/dusk differences). The models and observations should be consistent with one another (i.e., in the same coordinate system). Please provide additional description.*

> As suggested by the reviewer, we checked that the models were initially fitted in aberrated

460    coordinates. It appears that both models used these coordinates to obtain their fitting parameters.  So this should not be an issue in causing the dawn-dusk differences. We have included a brief discussion of this issue in the update manuscript when presenting Figure 5.

465    *There are several sentences in Sections 3 and 4 which are too convoluted and ambiguous for the reader to understand. This (along with several other issues described in this report) strongly suggests that the co-authors have not read this manuscript. A few examples include Lines 264-269; Lines 332-334; Lines 371-372.*

        We agree with the reviewers on the convoluted sentences and have rephrased them in an
470    updated manuscript to simplify the sentence structure and make the meaning clearer.

**Additional Minor Points of Referee #2:**

*Line 377: 'the foreshock develops directly in front of the bow shock nose,'. This is not strictly true. The point of 'attachment' of the foreshock to the bow shock surface is not necessarily the nose of*
475    *the bow shock; but wherever the IMF is tangent to the bow shock surface. Please reword.*

        We have changed the part "directly in front of the bow shock nose" to a more general statement. However, shouldn't in the context of the quasi-radial IMF, the position of the foreshock be directly in front of the nose of the bow shock, as this is where the IMF is most radial on the bow shock surface allowing the backstreaming of ions?

480    *Lines 408-409: 'the EJC plasma composition': What does this mean? Is this the general characteristics of Table 1? Or does this refer to a higher percentage of heavy ions (e.g., alphas)? Or does this refer to a composition that includes a higher than normal 'hot' plasma content? Please be specific.*

        Plasma composition refers here to the characteristics shown in Table 1. We have added
485    more information in our updated manuscript to the sentence to make this clear.

 *Lines 433-434: '(SMILE) mission … will again be a near-Earth, polar-orbiting satellite,'. This isn't right. The SMILE mission is rather high inclination (70° or 98°); but not a polar mission (inclination of 90°). It's also not in a near-Earth orbit: It's to go into a highly-elliptical orbit with perigee of ~1.8 RE (5000 km altitude), and apogee of ~20 RE.*

490    To avoid confusion with our wording of the polar mission, we have rephrased the sentence in our manuscript. The new wording is more in line with the information on the ESA SMILE website and the information provided by the reviewer.

*Lines 453-454: As described earlier, the Song et al. 1988 study examined the amplitude of MP motion, and the Russell et al. 1997 study already examined this amplitude in relation to the*
495    *foreshock (IMF orientation and local time). How would this future study be different?*

        Since our data set is much larger than that used in the aforementioned studies, we can study the amplitude changes of the MP motion due to the foreshock over multiple solar cycles, which to our knowledge hasn't been done extensively. In particular, the results

extracted from the very weak last solar cycle may differ from those published previously.

500     We have extended our conclusion to take this information into account.